# ARTICLES

## nature
## ecology & evolution
# Global patterns and rates of habitat transitions across the eukaryotic tree of life

Mahwash Jamy [1], Charlie Biwer[1], Daniel Vaulot [2], Aleix Obiol [3], Hongmei Jing[4], Sari Peura[5,6], Ramon Massana [3] and Fabien Burki [1,6 ✉]

The successful colonization of new habitats has played a fundamental role during the evolution of life. Salinity is one of the strongest barriers for organisms to cross, which has resulted in the evolution of distinct marine and non-marine (including both freshwater and soil) communities. Although microbes represent by far the vast majority of eukaryote diversity, the role of the salt barrier in shaping the diversity across the eukaryotic tree is poorly known. Traditional views suggest rare and ancient marine/non-marine transitions but this view is being challenged by the discovery of several recently transitioned lineages. Here, we investigate habitat evolution across the tree of eukaryotes using a unique set of taxon-rich phylogenies inferred from a combination of long-read and short-read environmental metabarcoding data spanning the ribosomal DNA operon. Our results show that, overall, marine and non-marine microbial communities are phylogenetically distinct but transitions have occurred in both directions in almost all major eukaryotic lineages, with hundreds of transition events detected. Some groups have experienced relatively high rates of transitions, most notably fungi for which crossing the salt barrier has probably been an important aspect of their successful diversification. At the deepest phylogenetic levels, ancestral habitat reconstruction analyses suggest that eukaryotes may have first evolved in non-marine habitats and that the two largest known eukaryotic assemblages (TSAR and Amorphea) arose in different habitats. Overall, our findings indicate that the salt barrier has played an important role during eukaryote evolution and provide a global perspective on habitat transitions in this domain of life.

Adapting to new environments with very different conditions represents large evolutionary steps. When successful, habitat transitions can be important drivers of evolution and trigger radiations[1–4]. The marine/non-marine boundary[5,6]—the so-called salt barrier—is considered one of the most difficult barriers to cross because salinity preference is a complex trait that requires the evolution of multigene pathways for physiological adaptations[7–10]. These adaptations have been best studied in macro-organisms, for which the recorded marine/non-marine transitions are few[11–13]. Microbes (prokaryotic and eukaryotic) are also typically regarded as infrequently crossing the salt barrier in spite of much larger population sizes and high dispersal ability[12,14] but the role of this barrier as an evolutionary driver of microbial diversity remains poorly understood. For bacteria, higher habitat transition rates than anticipated have been reported[15]. For microbial eukaryotes, which represent the vast majority of eukaryotic diversity, no data exist to infer the global patterns and rates of habitat transitions at a broad phylogenetic scale. Extant marine and non-marine eukaryotic communities (here, non-marine encompasses both freshwater and soil[5,6]) are distinct in terms of composition and abundance of taxa[6,16], a pattern that has been attributed to rare and ancient transitions between these two fundamentally different environments[14,17–22]. However, an increasing number of inferences of recent transitions in specific clades such as dinoflagellates suggest that the salt barrier might not be as strong as previously envisioned[23–26].

In this study, we used a unique hybrid approach combining high-throughput long-read and short-read environmental sequencing to infer habitat evolution across the eukaryotic tree of life. We newly generated over 10 million long environmental reads (~4,500 base pairs (bp)) of the ribosomal DNA (rDNA) operon) from 21 samples spanning marine (including the euphotic and aphotic ocean layers), freshwater and soil habitats. The increased phylogenetic signal of long-reads allowed us to establish, together with a set of phylogenomic constraints, a broad evolutionary framework for the environmental diversity of eukaryotes. We then incorporated existing, massive short-read data (~234 million reads) from a multitude of locations around the world to complement the taxonomic and habitat diversity of our dataset. With this combined dataset, we inferred the frequency, direction and relative timing of marine/non-marine transitions during the evolution of eukaryotes; we investigated which eukaryotic lineages are more adept at crossing the salt barrier; and finally, we reconstructed the most likely ancestral habitats throughout eukaryote evolution, from the root of the tree to the origin of all major eukaryotic lineages. Our analyses represent a comprehensive attempt to leverage environmental sequencing to infer the evolutionary history of habitat transitions across eukaryotes.

## Results

**Long-read metabarcoding to obtain a comprehensive phylogeny of environmental diversity.** A range of samples collected from marine and non-marine habitats were deeply sequenced with PacBio (Sequel II) to obtain a comprehensive long-read metabarcoding dataset spanning the broad phylogenetic diversity of eukaryotes. These samples covered several major ecosystems, including the marine euphotic and aphotic zones (surface/deep chlorophyll

[1]Department of Organismal Biology (Systematic Biology), Uppsala University, Uppsala, Sweden. [2]CNRS, UMR7144, Team ECOMAP, Station Biologique, Sorbonne Université, Roscoff, France. [3]Department of Marine Biology and Oceanography, Institut de Ciències del Mar (ICM-CSIC), Barcelona, Spain. [4]CAS Key Lab for Experimental Study Under Deep-sea Extreme Conditions, Institute of Deep-sea Science and Engineering, Chinese Academy of Sciences, Sanya, China. [5]Department of Ecology and Genetics (Limnology), Uppsala University, Uppsala, Sweden. [6]Science for Life Laboratory, Uppsala University, Uppsala, Sweden. ✉e-mail: fabien.burki@ebc.uu.se

maximum and mesopelagic/bathypelagic, respectively), freshwater lakes and ponds as well as tropical and boreal forest soils (Supplementary Table 1). In total, we obtained 10.7 million circular consensus sequence (CCS) reads spanning ~4,500 bp of the rDNA operon, from the 18S to the 28S rDNA genes. After processing, sequences were clustered into operational taxonomic units (OTUs) within each sample at 97% similarity, resulting in 16,821 high-quality OTUs. To assess the potential amplification and sequencing biases of long-read metabarcoding, we performed a direct comparison with Illumina data (for the V4 and V9 hypervariable regions of the rDNA gene and 18S reads extracted from metagenomic data) previously obtained for the same DNA from three marine samples[27]. This comparison revealed that our long-range PCR assay followed by PacBio sequencing retrieved relatively similar eukaryotic community snapshots. Most groups were detected at comparable abundances, with the exception of the MALV-I group that was detected at greater abundances with the long-read approach (Extended Data Figs. 1 and 2). The PacBio datasets also contained several taxonomic groups, such as diplonemids, kinetoplastids and MAST-25, that are absent from the V4 or V9 datasets. Importantly, over 80% of the V4 sequences were identical to the PacBio OTUs, indicating that our protocol for CCS processing generates high-fidelity data comparable to classical short-read metabarcoding (Extended Data Fig. 1).

All PacBio OTUs were labelled with appropriate taxonomic information using a phylogeny-aware method[28] (Methods) and used to reconstruct a eukaryotic phylogeny of environmental diversity (Fig. 1, Supplementary Fig. 1 and Supplementary Note 1). We refer to this phylogeny as the global long-read eukaryotic phylogeny as it contains almost all known major eukaryotic lineages (Fig. 1); the main missing groups represent large multicellular organisms or protists found in specific environments not sampled here (for example, anoxic environments; Supplementary Table 2). We also uncovered a proportion of novel diversity; that is, OTUs highly dissimilar to reference sequences that are typically difficult to confidently assign to taxonomic groups. Long-read metabarcoding alleviates the issue of taxonomic assignment of highly diverging sequences; for example, we found 863 sequences with <85% similarity to references in the protist ribosomal reference (PR²) database that were attributed a taxonomy on the basis of their position in the tree, mostly belonging to apicomplexan parasites, fungi and amoebozoans (Fig. 1a and Extended Data Fig. 3). To allow for transition rate calculations within a guiding taxonomic framework (see later), the major eukaryotic groups shown in Fig. 1a were constrained to be monophyletic on the basis of established relationships derived from phylogenomic inferences (reviewed in ref. [29]). These major lineages were defined as rank 4 in the taxonomic scheme of an in-house database derived from the PR² database[30] called PR2-transitions[31].

**Detection of a salty divide in microbial eukaryotes.** The global phylogeny in Fig. 1 shows habitat preferences across the eukaryotic tree of life. Overall, we observed a clear phylogenetic distinction between marine and non-marine lineages, with almost no OTU overlap between these two communities (Fig. 1b,c; Unifrac distance = 0.959, $P < 0.001$). Within each side of the salt barrier, soil and freshwater communities were found to be more distinct from each other (Unifrac distance = 0.76, $P < 0.001$) than the marine euphotic and aphotic communities (Unifrac distance = 0.64, $P < 0.001$) (Fig. 1b and Supplementary Fig. 2). However, we detected several sequences with high identity (>97% similar) present in the marine euphotic and aphotic samples (854 OTUs) and in the soil and freshwater samples (771 OTUs), suggesting that some taxa may be generalists in these subhabitats (Fig. 1c and Supplementary Fig. 3).

We next sought to increase the number of samples and diversity by taking advantage of the massive available short-read metabarcoding datasets. We gathered data from 22 studies conducted globally (including marine and non-marine ecosystems), amounting to 234 million reads in total after processing (Supplementary Fig. 4 and Supplementary Table 3). We opted to use only the V4 region (~380 bp) of the 18S rDNA gene as it was shown to have a greater phylogenetic signal than the V9 region[32]. The V4 reads were clustered into OTUs at 97% similarity for the marine euphotic (9,977 OTUs), marine aphotic (2,518 OTUs), freshwater (3,788 OTUs) and soil (11,935 OTUs) environments (Supplementary Table 4). These short-read OTUs were then phylogenetically placed onto the global long-read eukaryotic phylogeny using the evolutionary placement algorithm (EPA)[33] (Extended Data Fig. 4), for which we compared the placement distributions for each subhabitat. Interestingly, most placements occurred close to the tips of the reference tree, indicating that our long-read dataset adequately represents the diversity recovered by short-read metabarcoding (Extended Data Fig. 4). Furthermore, the placement distributions for each habitat are consistent with our results based on the long-reads only, namely that marine and non-marine communities are distinct and, at a finer level, soil and freshwater communities are more different from each other than communities in the surface and deep ocean (soil–freshwater earth mover's distance = 1.14, marine euphotic–aphotic earth mover's distance = 0.809; Supplementary Fig. 5).

**Habitat transition rates vary across major eukaryotic clades.** The above results confirm that the salt barrier leads to phylogenetically distinct eukaryotic communities. We next asked (1) how often have transitions between marine and non-marine habitats occurred during evolution, (2) which eukaryotic lineages have crossed this barrier more frequently and (3) in which direction? To answer these questions, we calculated habitat transition rates across the global eukaryotic phylogeny by performing Bayesian ancestral state reconstructions using continuous-time Markov models[34]. We first tested a homogenous model, where a single pattern of transition rates from marine to non-marine habitats ($q_{M-NM}$) and vice versa ($q_{NM-M}$) was estimated across all eukaryotes. The homogenous model returned a posterior density of log-likelihoods with a mean of −2,008.45 and transitions from marine to non-marine habitats were found to be just as likely as the opposite direction across the tree ($q_{M-NM} = q_{NM-M} = 0.19$ transitions per substitution per site; Extended Data Fig. 5). However, the assumption of the homogenous model of uniform transition rates across the tree may be violated if there are large variations in habitat transition rates between groups. Indeed, a heterogenous model where we estimated $q_{M-NM}$ and $q_{NM-M}$

**Fig. 1 | Global eukaryotic 18S–28S phylogeny from environmental samples and the distribution of habitats. a**, This tree corresponds to the best maximum-likelihood tree inferred using an alignment with 7,160 sites and the GTRCAT model in RAxML[99]. The tree contains 16,821 OTUs generated from PacBio sequencing of 21 environmental samples (no reference sequences were included). Ring no. 1 around the tree indicates taxonomy of the environmental sequences, with all major eukaryotic lineages considered in this study labelled. Ring no. 2 depicts percentage similarity with the references in the PR² database as calculated using BLAST and was set with a minimum of 70% with the two black lines in the middle indicating 85% and 100% similarity levels. Ring no. 3 depicts the habitat origin of each OTU. **b**, Hierarchical clustering of the four habitats based on a phylogenetic distance matrix generated using the unweighted UniFrac method ($n = 7$, $n = 5$, $n = 4$ and $n = 5$ samples for soil, freshwater, marine euphotic and marine aphotic, respectively). All communities were found to differ significantly from each other using Monte Carlo simulations (Bonferroni-adjusted $P < 0.001$). **c**, Stacked density plot of branch lengths between taxa pairs from the same or different habitats ($n = 14,977,604$ taxa pairs with a maximum patristic distance of 1.5 substitutions/site). Note that this plot should be interpreted with caution as taxa pairs do not represent independent datapoints due to phylogenetic relatedness.

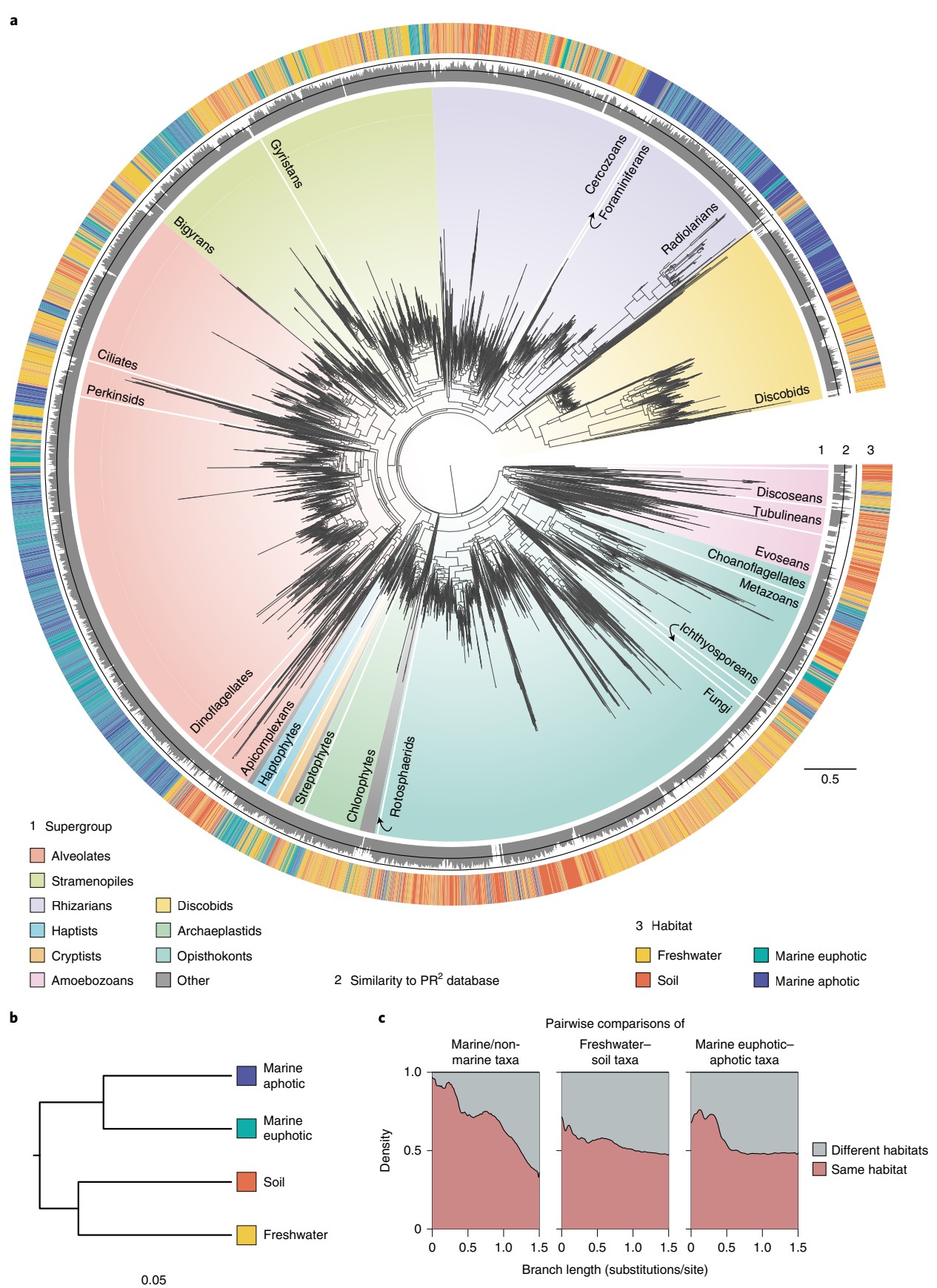

**a**

Gyristans

Bigyrans

Ciliates

Perkinsids

Cercozoans

Foraminiferans

Radiolarians

Discobids

Dinoflagellates

Apicomplexans

Haptophytes

Streptophytes

Chlorophytes

Rotosphaerids

1 2 3

Discoseans

Tubulineans

Evoseans

Choanoflagellates

Metazoans

Ichthyosporeans

Fungi

0.5

**1 Supergroup**

- Alveolates
- Stramenopiles
- Rhizarians
- Haptists
- Cryptists
- Amoebozoans
- Discobids
- Archaeplastids
- Opisthokonts
- Other

**2 Similarity to PR² database**

**3 Habitat**

- Freshwater
- Soil
- Marine euphotic
- Marine aphotic

**b**

- Marine aphotic
- Marine euphotic
- Soil
- Freshwater

0.05

**c** Pairwise comparisons of

Marine/non-marine taxa · Freshwater–soil taxa · Marine euphotic–aphotic taxa

Density

Branch length (substitutions/site)

- Different habitats
- Same habitat

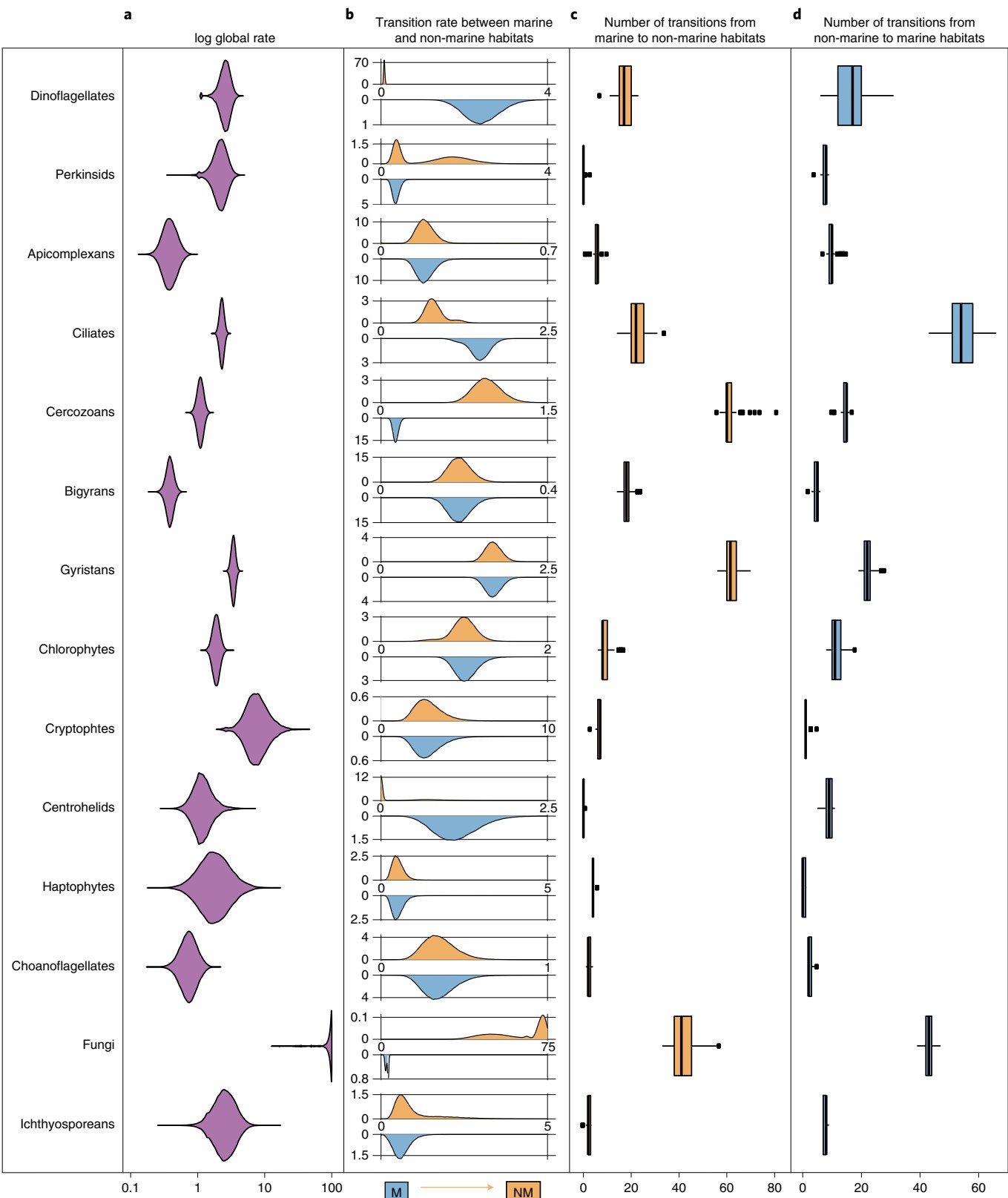

**Fig. 2 | Habitat transition rates and number of transition events estimated for each major eukaryotic lineage. a**, Posterior probability distributions of the global rate of habitat evolution, which indicate the overall speed at which transitions between marine and non-marine habitats have occurred in each clade regardless of direction. Rates were estimated along clade-specific phylogenies (Extended Data Fig. 6) using MCMC in BayesTraits with a normalized transition matrix. **b**, The posterior probability distribution of transition rates from marine to non-marine habitats (top in orange) and from non-marine to marine habitats (below in blue). **c,d**, Number of transitions from marine to non-marine habitats (**c**) and in the reverse direction (**d**) for each clade as estimated by PASTML using maximum likelihood (Methods). The boxplots in **c** and **d** show the median as centre line, box sizes indicate the lower (Q1) and upper (Q3) quartiles, whiskers indicate extreme values within 1.5× the interquartile range and dots beyond the whiskers indicate outliers.

separately for each major eukaryotic clade, presented a much better fit (log-likelihood score of −1,819.91; log Bayes factor = 269.3; Extended Data Fig. 5), indicating that habitat transition rates vary strongly across the tree.

To investigate in more detail the rate of habitat transition within each major eukaryotic group, we inferred taxon-rich clade-specific phylogenies by combining short-read data with the backbone phylogenies obtained from long-read data (with average SH-like support values varying between 72 and 85). Incorporating these short-read data allowed us to detect additional transition events that would have otherwise been missed with the long-read data alone (Extended Data Fig. 6). We modelled habitat transition rates along clade-specific phylogenies containing both marine and non-marine taxa that were sufficiently large (at least 50 tips) to get precise estimates. We also excluded discobid excavates and discosean amoebozoans because preliminary analyses showed ambiguous transition rate estimates owing to large phylogenetic uncertainty. Fungi were found to have by far the highest number of transitions per unit of evolutionary change; we estimated around 90 expected transition events along a branch length of one substitution/site (but see Supplementary Note 2). These results indicate that habitat shifts are associated with very little evolutionary change in the rDNA sequences (Fig. 2a). After fungi, cryptophytes and gyristans (ochrophyte algae, oomycete parasites and several free-living flagellates) had the highest global rates (around 8.2 and 3.4 expected transitions per substitution per site).

At a finer phylogenetic resolution, several subclades within stramenopiles, ciliates and dinoflagellates, seem particularly adept at crossing the salt barrier, especially chrysophytes, diatoms and spirotrich ciliates (11.8, 8.7 and 3.8 expected transition events per substitution per site respectively; Extended Data Fig. 7 and Supplementary Figs. 6 and 7). At the other extreme, groups such as bigyrans (heterotrophic stramenopiles related to gyristans) and apicomplexans (a group of parasites including the malaria pathogen) displayed the lowest habitat transition rates (around 0.4 expected transitions for every substitution per site). These results were further confirmed with sequence similarity network analyses, which showed high assortativity between marine and non-marine sequences for bigyrans and apicomplexans (meaning that non-marine and marine sequences formed distinct clusters at varying similarity thresholds), as opposed to gyristans and fungi, which showed low assortativity (Supplementary Fig. 8).

Within each major eukaryotic group, we next inferred the frequency for each direction of the transitions between marine to non-marine habitats. We found that all clades investigated had non-null transition rates in both directions, with the exception of centrohelids for which a model with a non-marine colonization rate set to zero was sampled 73% of the time (Fig. 2b). These results indicate that in nearly all major eukaryotic lineages containing non-marine and marine taxa, transitions have occurred in both directions. Some clades had symmetrical transition rates, indicating that the tendency to colonize marine environments was not significantly different from the tendency to colonize non-marine environments; this was, for example, the case for apicomplexans, bigyrans, chlorophytes, cryptophytes, haptophytes and choanoflagellates (Fig. 2b). However, some groups showed marked directionality preferences. Dinoflagellates, for example, show a much greater transition rate for colonizing marine habitats (about 31 times more likely). On the other hand, transitions to non-marine environments were significantly more likely than the reverse direction for fungi and cercozoans (about 21.5 and 7.2 times more likely, respectively). These trends in directionality were largely robust to variations in sampling efforts with the exception of ciliates, where subsampling marine euphotic taxa resulted in symmetrical transition rates and gyristans, for which subsampling marine euphotic taxa changed transition patterns from symmetrical to asymmetrical (towards non-marine habitats) in some cases (Supplementary Fig. 9). Finally, the directionality

of habitat transition appears to be heterogeneous also within the major eukaryotic groups (Supplementary Figs. 10–13). Indeed, for some selected subclades such as ascomycetes and basidiomycetes within fungi, the transition rates to marine environments were higher as compared to non-Dikarya fungi ($q_{NM-M}=8.47$ versus 1.65, respectively; Supplementary Fig. 13), although fungi as a whole showed a marked tendency to colonize non-marine habitats.

Finally, we estimated the number of transition events within each clade by generating discrete habitat histories using a maximum-likelihood method[35]. We conservatively counted transition events only if they led to a clade with at least two taxa in the new habitat to distinguish between biologically active, speciating residents from wind-blown cells, resting spores or extracellular DNA from dead cells[36]. Our analyses revealed at least 350 transition events occurring over eukaryotic history, although the actual number is likely to be higher when considering lineages that have gone extinct. Out of these, 72 or more transition events occurred in fungi alone (39–47 transitions to marine environments detected and 33–57 transitions to non-marine environments detected) (Fig. 2c,d). This was closely followed by gyristans and ciliates, with >60 putative switches each between environments (Fig. 2c,d).

**Relative timing of habitat transitions.** We wanted to determine when during eukaryote evolution the transitions between marine and non-marine habitats occurred. To calculate a relative timing for all transitions, we converted the clade-specific phylogenies into chronograms with relative dates (as in ref. [37]).

For each putative transition event, we measured the relative branch length from the inferred transition to the root of the clade. The general trend is that most transitions occurred relatively recently in the history of the groups (Fig. 3). For instance, we detected no transition events in fungi older than 25% of the clade's history, with most transitions occurring in the last 10% of the time that this group has been on Earth. Assuming that fungi arose around 1 billion years ago[38–40], this would imply that >90% of all marine/non-marine transitions (at least 63 transitions according to our analyses) in fungi occurred in the last 100 million years alone, with older transitions occurring predominantly towards marine environments. The observation that most transitions occurred towards the present could be due to the increased challenges of inferring transition events early in the evolution of a group because of poorer resolution of deeper nodes due to little phylogenetic signal and/or unsuccessful transitions leading to lineage extinctions in the new habitat. However, for a few clades at least (centrohelids, bigyrans, apicomplexans, cercozoans and chlorophytes), we detected several early transitions in the evolution of the group (Fig. 3). Interestingly, the direction of these early habitat transitions is non-overlapping. For centrohelids and apicomplexans, the early transitions were mainly towards marine environments, possibly corresponding to repeated marine colonization events at the onset of the groups' evolution. Early non-marine colonization events were instead detected in cercozoans, chlorophytes and bigyrans. Altogether, these observations suggest that early in the evolution of the major eukaryotic groups the pressure to move towards marine or non-marine habitats was group-specific and directional.

**Ancestral habitat reconstruction.** Our global long-read eukaryotic phylogeny, combined with the clade-specific phylogenies including short-read metabarcoding data, represent a very dense set of environmental information put in a phylogenetic framework. We used this information to reconstruct in a Bayesian analysis the most likely ancestral environments from the root of the eukaryotic tree through the emergence of the major groups. Inferring the ancestral habitat of the last eukaryotic common ancestor (LECA) requires information about the root itself, which remains very contentious[29,41]. To accommodate uncertainties for the position of the root, we performed

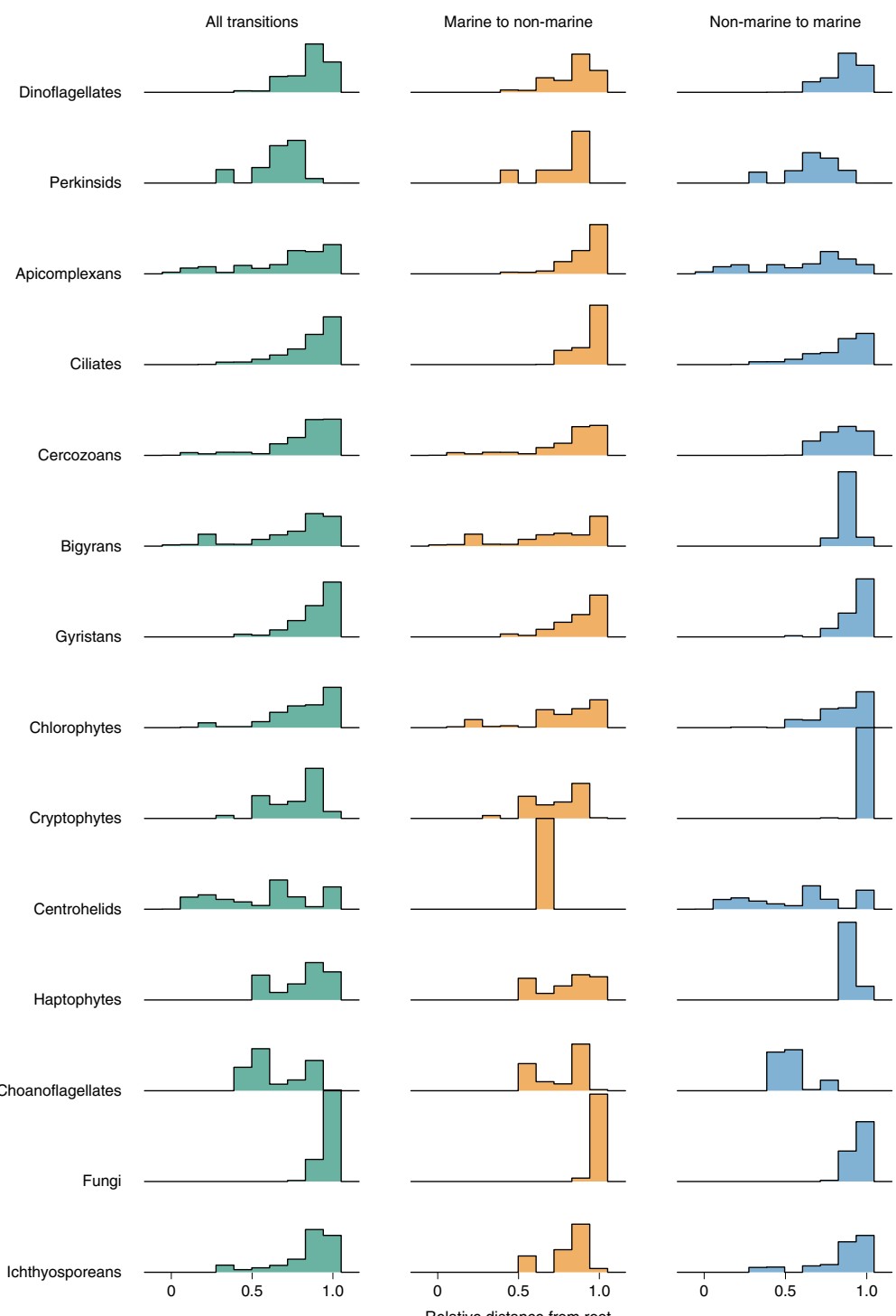

**Fig. 3 | Ridgeline histogram plots displaying the timing of transition events.** The plots were estimated from relative chronograms obtained with Pathd8 (ref. [37]). The *x* axis depicts the relative age for each clade.

ancestral habitat reconstruction analyses using the two most commonly proposed root positions: (1) between the discobid excavates and all other eukaryotes[42] and (2) between amorpheans (the group including animals, fungi and amoebozoans) and all other eukaryotes[43]. Both root alternatives converged towards the same habitats, suggesting that LECA evolved in a non-marine environment (Fig. 4a).

From the inferred non-marine root, our analyses suggest that two of the largest mega-assemblages of eukaryotes, probably comprising more than half of all eukaryotic diversity[44], arose in different environments. On one hand, the amorphean group probably originated in a freshwater or soil habitat (Fig. 4b), where it initially diversified into obazoans (which include well-known lineages such as animals and fungi but also several unicellular related lineages), as well as the amoebozoans. Consistent with previous studies, we inferred a marine origin for metazoans[45,46]; however, for two obazoan lineages—fungi and the group containing metazoans and choanoflagellates—we could not determine a clear preference for their ancestral habitats. On the other hand, our analyses indicate that

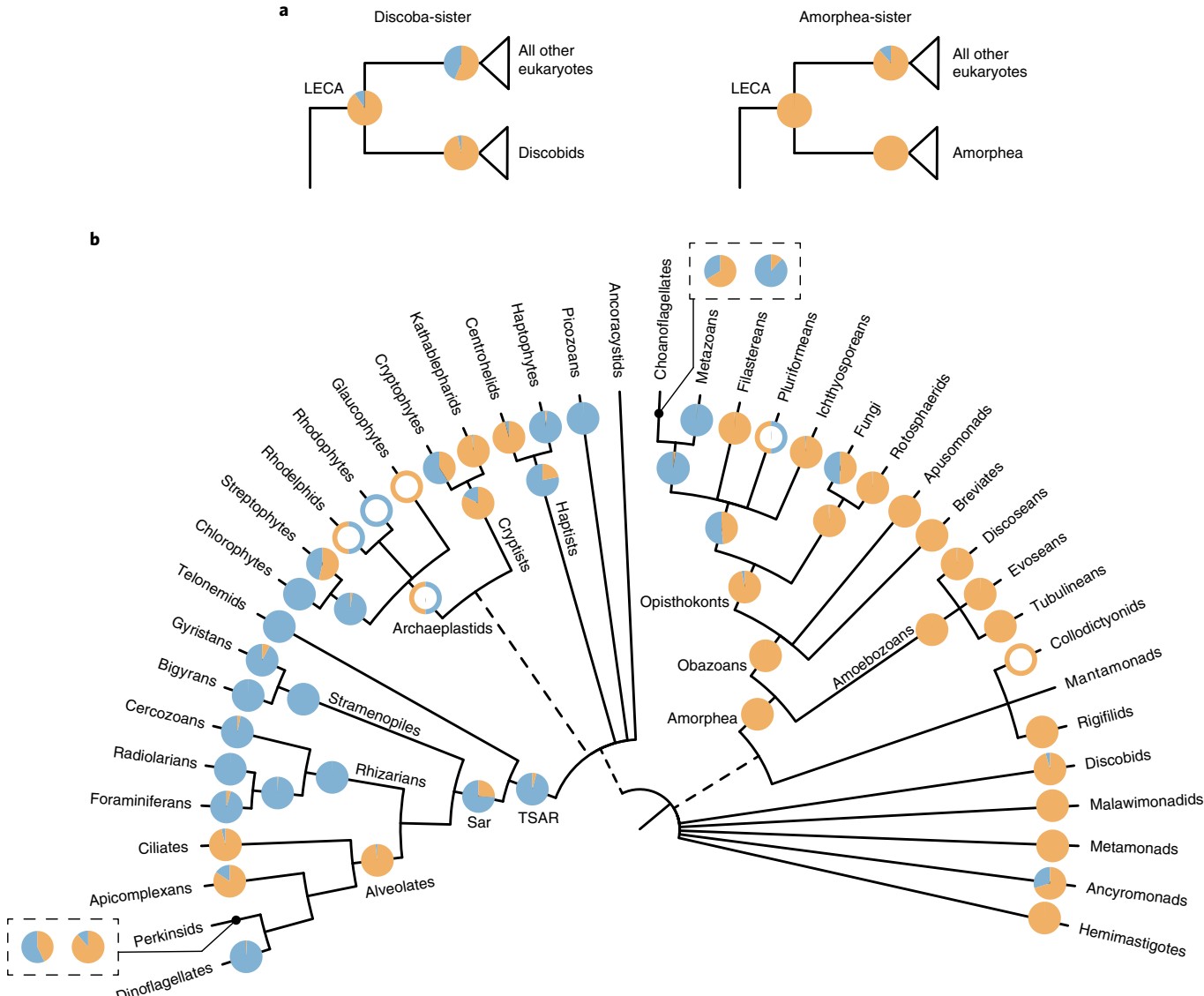

**Fig. 4 | Ancestral states of major eukaryotic clades as estimated by BayesTraits on a set of 100 global PacBio phylogenies.** Pie charts at each node indicate the posterior probabilities of likelihoods for the character states as follows: blue, marine; orange, non-marine. Nodes with empty circles indicate wherever there was insufficient taxon sampling to infer ancestral habitats but a reasonable estimate could be made from existing literature (Supplementary Note 3). **a**, Ancestral habitat of the LECA as inferred using two different roots. **b**, Ancestral states of major eukaryotic lineages. For the two cases where the incorporation of Illumina data inferred a different likely ancestral state, the results are shown in boxes. The pie chart on the right was obtained using the global eukaryotic phylogeny, while the pie chart on the left was obtained from clade-specific phylogenies. The tree is adapted from ref.[29].

the expansive TSAR clade (containing the main eukaryotic phyla stramenopiles, alveoates and rhizarians, as well as the smaller group telonemids) probably originated in a marine environment, following the transition of an ancestral population from a non-marine root (Fig. 4b). A marine origin is also likely for the major TSAR members, except for alveolates which were inferred to have a non-marine origin.

Overall, the predicted ancestral habitats of most major eukaryotic clades match their current preferred habitat: this is, for example, the case for all amoebozoan lineages, radiolarians, dinoflagellates and foraminiferans. An exception is cercozoans for which a marine origin was inferred but which now dominate non-marine environments, particularly soils[6,47]. Interestingly, the results derived from the global long-read eukaryotic phylogeny and the clade-specific phylogenies (which include short-read OTUs) were largely consistent, except in two cases: the phylogeny of perkinsids changed the

origin from non-marine to marine for these parasites of animals, while the phylogeny of choanoflagellates switched from a marine to a non-marine origin (Fig. 4b).

## Discussion

In this study, we used a unique combination of long- and short-read data to obtain an evolutionary framework of environmental diversity and infer habitat-preference evolution across the eukaryotic tree. High-throughput long-amplicon sequencing followed by careful processing of the data provide high-quality sequences containing improved phylogenetic signal for the vast environmental diversity[28,48–50]. We generated over 10 million long-read metabarcoding data spanning the eukaryotic rDNA operon, which assembled into nearly 17,000 OTUs, for marine and non-marine ecosystems. We then added two additional layers of phylogenetic information: (1) a much larger amount of available short-read metabarcoding

data to more deeply cover the molecular diversity of environmental microbes and (2) a set of well-accepted constraints derived from published phylogenomic analyses to fix the backbone of our eukaryotic tree. By combining all this information, we show that we can infer evolutionary patterns at global scales across the tree.

We confirm that the salt barrier has been a major factor in shaping eukaryotic evolution[6,14,16] and that marine/non-marine transitions are infrequent in comparison to transitions across other habitats such as between freshwater and soil (Fig. 1). Our analyses detected at least 350 transition events (Fig. 2), although this number is probably a minimum estimate when considering that: (1) extinct lineages missing from our phylogenies hide transitions; (2) future sampling efforts from more diverse geographical locations (for example, in ref. [24]) will help to detect more transitions; and (3) by clustering sequences into OTUs at 97% similarity, we do not detect recent transition events within the clusters (for example, in refs. [26,51,52]). These difficult-to-achieve environmental crossings have probably played important evolutionary roles by allowing colonizers to reach vacant ecological niches. Crossing the salt barrier may have led to the establishment of some major eukaryotic assemblages such as TSAR or highly diverse lineages such as the oomycetes and vampyrellids (Supplementary Fig. 7). Marine/non-marine transitions have also allowed lineages such as diatoms, golden algae and spirotrich ciliates to expand their range to both habitats, contributing to the diversification of the vast eukaryotic diversity we see today. We unexpectedly found that 56% of all detected transitions occurred recently, in the last 10% of the evolutionary history of the respective groups (Fig. 3), which is in contrast to a common idea that most marine/non-marine transitions are ancient[14]. It is, however, unclear why colonization across the salt barrier would be more frequent in recent geological time, so this observation could instead be due to recent colonizing lineages having had less time to go extinct and thus being more likely to be represented in our data[53].

At its deepest phylogenetic level, our analyses suggest that the earliest eukaryotes inhabited non-marine habitats (Fig. 4) and not marine habitats as often assumed (for example, refs. [54–56]). While the fossil record for early eukaryotes is sparse and difficult to distinguish from prokaryotes, there is evidence for early eukaryotes in non-marine or low-salinity environments from at least 1 billion years ago[55]. Furthermore, other key early eukaryotic innovations, such as the origin of the plastid organelles, have been inferred to have occurred around 2 billion years ago in low-salinity habitats[57,58]. Freshwater and soil environments are known to be more heterogeneous[59] and may thus have provided a wider range of ecological niches for early eukaryotes to occupy. However, one source of uncertainties in our ancestral state reconstructions is the lack of samples from some major ecosystems such as marine sediments, anoxic habitats and other extreme habitats like hydrothermal vents and hypersaline lakes. These habitats may contain deep-branching lineages not represented in our current phylogenies. Because ancestral state reconstruction analyses are especially sensitive to deep-branching lineages, the inclusion of environmental data from more habitat types (and the discovery of more kingdom-level lineages) is crucial to confirm the inferences presented here.

Our detailed investigation across the main groups of eukaryotes showed marked differences in the rates of crossing the salt barrier (200-fold globally). While some groups have low global transition rates, others show a higher tendency to cross this physiological barrier. Most notably, we inferred on the basis of both the highest transition rates in our analysis and relatively high number of transition events (Fig. 2), that fungi are the strongest eukaryotic colonizers between marine and non-marine environments. This is consistent with previous studies documenting a multitude of close evolutionary associations between marine and non-marine fungal lineages[60–62], which in turn suggests that many fungal species may be generalists that can tolerate a wide range of salinities[63,64]. Interestingly, fungi

showed a much greater trend (21-fold) for colonizing non-marine environments, where they are dominant, than the reverse. Whether this reflects a strong preference for non-marine environments or instead unequal diversification rates in the two habitats[65,66], or both, is unclear and should be further investigated.

The differences in habitat transition rates across eukaryotes are probably shaped by a host of complex factors. Microbial groups with lower dispersal ability, such as the large radiolarians and testate amoebae, probably have a lower number of potential colonizers, decreasing transition opportunities. Trophic lifestyles may also play a role, with parasites and symbionts such as apicomplexans and perkinsids potentially expected to mirror the habitat transition histories of their respective hosts. Furthermore, varying salinity tolerance can also prevent, or instead promote, successful colonization events. Among algae, comparative genomics showed large differences in gene content between marine and freshwater species, notably for ion transporters and other membrane proteins that probably play important roles in osmoregulation[67]. These different gene contents may be due, at least in part, to lateral gene transfers (LGT) that could facilitate successful crossing of the salt barrier, as proposed for other environmental adaptations[68–73]. Interestingly, for two of the most frequently transitioning eukaryotic groups (diatoms and ascomycetes), it has been shown that gene transfer is an important driver of evolution[74,75]. However, the precise role of LGT in facilitating the crossing of the salt barrier remains unclear, as is our understanding of how protists in general acquire the necessary genes for osmoregulation. Whether it is through LGT (as has been shown for some halophilic protists[68]), gene duplication or rewiring of existing metabolic pathways (as shown for the SAR11 bacteria[76]) remains to be studied. Other ecological factors must also intervene, as a colonizing organism does not only need to adapt to a different salinity but also has to adapt to the different nutrient and ion availabilities and avoid being out-competed or preyed upon by the resident community[73].

## Conclusions

This study represents the first comprehensive analysis of the evolution of marine and non-marine habitat preferences across the global tree of eukaryotes. We inferred that two of the largest assemblages of eukaryotes (TSAR and Amorphea) originated in different environments and that ancestral eukaryotes probably inhabited non-marine environments. Our results show that marine and non-marine communities are phylogenetically distinct but the salt barrier has been crossed at least several hundred times over the course of eukaryotic evolution. Several of these crossings coincided with the birth of diverse lineages, indicating that the availability of new niches has probably played a large role in the vast eukaryotic diversity we see today. We predict that the generation of genomic data from closely related marine and non-marine lineages will shed light on the genetic and cellular adaptations that have allowed crossings over the salt barrier.

## Methods

**Environmental samples and total DNA extraction.** A total of 18 samples were sequenced for this study: five freshwater samples, four soil samples, four marine euphotic samples and five marine aphotic samples (see Supplementary Table 1 for sample coordinates and details). Additionally, we used reads from three soil samples that were sequenced in a previous study[28] (European Nucleotide Archive (ENA) accession PRJEB25197), resulting in a total of 21 samples that were analysed in this study. The aim here was to get a representative view of the microbial eukaryotic diversity in each environment using long-read metabarcoding.

*Soil samples (four).* Peat samples were collected from (1) Skogaryd mire and (2) Kallkäls mire in October–November 2019. The 5 ml samples with three to four replicates of the top layer of soil were collected at both sites and visible roots were removed. Samples were kept at 4 °C for 2 d before extracting DNA using the DNeasy PowerSoil Kit (Qiagen). We also obtained DNA extracts from: (3) rainforest soil samples (six sites) from Puerto Rico[77] and (4) boreal forest soil samples (six sites) from Sweden[78].

*Freshwater samples (five).* We sampled three freshwater lakes in Sweden in October–November 2019: (1) Lake Erken, (2) Lake Ersjön and (3) Lake Stortjärn. Planktonic samples were collected from the middle of the lakes at multiple depths and mixed. Up to 3 l of water was prefiltered through a 200 µm mesh net to remove larger organisms before sequential filtration through 20–25 µm, 3 µm and 0.25 µm polycarbonate filters (47 mm). Filters were immediately frozen at −20 °C and stored at −70 °C before further processing. We also collected a (4) freshwater sediment sample (four replicates) from Lake Erken. The upper 0–5 cm of a sediment core was separated and mixed. All samples were kept at 4 °C before processing and extracting DNA using the DNeasy PowerSoil Kit. Lastly, we obtained DNA from (5) ten permafrost thaw ponds in Canada[79].

*Marine euphotic samples (four).* One 5 l sample was collected from the (1) North Sea at a depth of 5 m. Water was processed and DNA extracted as described for the freshwater water samples. We used DNA extracts from the nano-plankton (3–20 µm) and pico-plankton (0.2–3 µm) fractions of two stations from the Malaspina expedition (stations 49 and 76 in the Indian Ocean)[80]. These extracts corresponded to (2) one surface sample at 3 m depth and (3 and 4) two deep chlorophyll maxima (DCM) layer samples at depths of 70 and 85 m.

*Marine aphotic samples (five).* We used DNA extracts from the nano- and pico-fractions of the aphotic marine environment from Malaspina stations 49 and 76 (ref. [80]). These corresponded to depths of (1 and 2) 275 and 800 m for the mesopelagic and (3 and 4) 1,200 and 2,800–3,300 m for the bathypelagic samples. Lastly, we obtained (5) DNA from a Mariana Trench sample from a depth of 5,900 m (ref. [81]).

*Dataset consistency.* The DNA from several samples (primarily non-Swedish samples) were obtained from other studies. The samples were processed and DNA extracted using distinct methods, which can introduce various biases in the overall community obtained. We assessed how robust the community profiles obtained were to (1) different extraction methods and (2) different filtration protocols.

Different commercial kits were used for extracting DNA from the environmental samples. Additionally, several extraction protocols (of two soil and seven marine samples) included extra cell-beating or cryogenic crushing steps[77,78,80]. While it has been previously shown that DNA extraction protocols significantly affect the protist communities retrieved, most of these differences are restricted to several groups and do not overwhelm the real, biological variations between samples[82]. Given the limited number of samples in this study, it is difficult to assess how much of the variation between samples can be attributed to the extraction protocol but differences are likely to be minor.

Freshwater and marine samples were filtered through meshes with different pore sizes. The smallest pore size did not differ in both cases (0.2–0.25 µm which enables capturing even the smallest eukaryotes such as *Ostreococcus tauri* with a diameter of 0.8 µm; ref. [83]). On the other hand, the largest pore size did differ between freshwater and marine samples: a 'micro' size fraction (20–200 µm) was obtained for most freshwater samples but not for most marine samples. However, we do not expect this difference to influence the communities obtained as most of the marine protist diversity is captured in the smaller size fractions (with the exception of the exclusively marine Collodaria and Phaeodarea which are mostly found in the micro-size fraction)[84]. Overall, we are confident that the aquatic communities obtained are comparable to the soil communities in this study.

**PCR amplification and long-read sequencing.** We amplified a ~4,500 bp fragment of the rDNA operon using the general eukaryotic primers 3NDf[85] and 21R[86], including part of the 18S gene, the complete internal transcribed spacer (ITS) region, and part of the 28S gene[85,86]. PCRs were performed with sample-specific tagged-primers using the Takara LA Taq polymerase (Takara) and 5 ng of DNA as input. PCR-cycling conditions included an initial denaturation step at 94 °C for 5 min, at least 25 cycles of denaturation at 98 °C for 10 s, primer annealing at 60 °C for 30 s and elongation at 68 °C for 5 min and finishing with a final elongation step at 68 °C for 10 min. We limited the number of PCR cycles to 25, where possible, to reduce chimaera formation[87]. For samples that did not get amplified, we increased the number of cycles to 30. PCR products were assessed using agarose gels and Qubit 2.0 (Life Technologies) and then purified with Ampure XP beads (Beckman Coulter). Amplicons from replicates, size fractions and different sites from the same sampling location were pooled at this stage. SMRTbell libraries were constructed using the HiFi SMRTbell Express Template Prep Kit 2.0. Long-read sequencing was carried out at SciLifeLab (Uppsala, Sweden) on the Sequel II instrument (Pacific Biosciences) on a SMRT Cell 8 M Tray (v.3), generating four 30-h movies.

**Processing reads and OTU clustering.** We QC filtered sequences following ref. [28] with some modifications. The CCS filtration pipeline is available on GitHub[88]. Briefly, CCSs were generated by SMRT Link v.8.0.0.79519 with default options. The CCS reads were demultiplexed with mothur v.1.39.5 (ref. [89]) and then filtered with DADA2 v.1.14.1 (ref. [90]). Reads were retained if they had both primers and if the maximum number of expected errors was four (roughly translating to one error for every 1,000 bp). We preclustered reads at 99% similarity using VSEARCH v.2.3.4 (ref. [91]) and generated consensus sequences for preclusters ≥3 reads to denoise

the data. Prokaryotic sequences were detected by BLASTing[92] against the SILVA SSU Ref NR 99 database v.132 (ref. [93]) and removed. We predicted 18S and 28S sequences in the reads using Barrnap v.0.9 (--reject 0.4 --kingdom euk) (https://github.com/tseemann/barrnap) and discarded non-specific and artefactual reads (those containing multiple 18S/28S or missing 18S/28S). Chimaeras were detected de novo using Uchime[94] as implemented in mothur. Finally, we extracted the 18S and 28S sequences from the reads and clustered them using VSEARCH into OTUs at 97% similarity. After discarding singletons, a second round of de novo chimaera detection was performed using VSEARCH and chimaeric OTUs were removed. We calculated sequence similarity of the OTUs against reference sequences in a custom PR² database[30] (PR2-transitions[31]; see later) using two methods. (1) A global identity search was carried out using VSEARCH (--usearch_global and --iddef 1; Extended Data Fig. 3). For this method, all references and OTU sequences were trimmed with the primers 3ndf and 1510R[95] using Cutadapt[96] to ensure that they spanned the same region. (2) Since not all sequences in PR² span the region between 3ndf and 1510R or are targeted by this primer pair, we also estimated local similarity by BLASTing the 18S OTU sequences against references in the PR² database and extracting the top hit with an alignment of at least 500 bp. The corresponding percentage identities are displayed in Fig. 1.

**Taxonomic annotation of long-read sequences.** *The modified PR² reference database.* Reference sequences were derived from a modified version of the Protist Ribosomal Reference (PR²) database v.4.12.0 (ref. [30]), called PR2_transitions. This database used a revised taxonomy structure compared to PR², with nine instead of eight levels adding a Subdivision level: Domain, Supergroup, Division, Subdivision, Class, Order, Family, Genus and Species. This allowed us to update the taxonomy to accommodate recent changes in eukaryotic classification[97] (changes in taxonomy can be viewed at ref. [88]). Additionally, we added sequences from nucleomorphs and several newly discovered or sequenced lineages such as Rholphea, Hemimastigophora and others. PR2_transitions is available on Figshare[31]. We used the 18S gene alone for taxonomic annotation, as 28S databases are much less comprehensive by comparison.

*Phylogeny-aware taxonomy assignment.* We used a phylogeny-aware approach to assign taxonomy to the PacBio OTUs, as done in ref. [28]. This approach assigns taxonomy to the appropriate taxonomic rank, such that OTUs branching deep in the eukaryotic tree are labelled to high taxonomic ranks and vice versa. For each sample, we inferred preliminary maximum-likelihood trees along with SH-like support[98] with RAxML v.8 (ref. [99]) (using the GTRCAT approximation as it is better suited for large trees[100]). These trees contained the filtered OTUs and closely related reference sequences from PR2_transitions. Trees were scanned manually to identify misannotated reference sequences, nucleomorphs and artefactual OTUs. After removing these sequences, we inferred trees with RAxML-NG[101] using 20 starting trees.

The final taxonomy was generated by getting the consensus of two strategies. Strategy 1 parses the tree and propagates taxonomy to the OTUs from the nearest reference sequences using the Genesis[102] app partial-tree-taxassign (https://github.com/Pbdas/genesis-apps/blob/master/partial-tree-taxassign.cpp). Strategy 2 starts by pruning the OTUs from the phylogeny, leaving behind references only. OTUs are then phylogenetically placed on the tree with EPA-ng v.0.3.5 (ref. [33]) and taxonomy assigned using the gappa[102] command assign under the module examine. The resulting taxonomy of the 18S gene of each OTU was transferred to its 28S gene counterpart, as the molecules are physically linked. The taxonomic annotations of the OTUs were used downstream to label clades in the global long-read phylogeny (Fig. 1) as well as to enforce monophyly of major eukaryotic lineages (see next section).

**Maximum-likelihood analyses of the global eukaryotic dataset.** The 18S and 28S sequences were aligned using MAFFT v.7.310 (ref. [103]) using the FFT-NS-2 strategy and subsequently trimmed with trimAl[104] to remove sites with >95% gaps. We inferred preliminary trees from a concatenated alignment with RAxML v.8.2.12 under the GTRCAT model[29] which were then visually inspected to detect chimaeras and sequence artefacts. Taxa were removed if their position in the tree did not match their taxonomy. Four such rounds of visual inspection were performed, two with unconstrained trees and two with constrained trees (see text below for details on constraints). To avoid long-branch attraction, we excluded rapidly evolving taxa using TreeShrink[105] ($k = 2,500$). This resulted in the removal of *Mesodinium*, long-branch Microsporidia, several Apicomplexa, several Heterolobosea and several Colladaria from our dataset.

After removing chimaeras and sequence artefacts, we realigned and trimmed the 18S and 28S sequences as before. After concatenation, the final dataset was composed of 16,821 taxa and 7,160 alignment sites. Global eukaryotic phylogenies of the taxonomically annotated, 18S–28S environmental sequences were inferred using RAxML v.8.2.12 under the GTRCAT model[99] and 100 transfer bootstrap replicates (TBE)[106]. Supergroups, Divisions and Subdivisions (ranks 2, 3 and 4 in PR2_transitions) were constrained to be monophyletic in our tree (all taxa labelled as a specific subdivision were constrained to be on one side of a split). The one exception was Excavata whose monophyly has not been confidently resolved[29]. One-hundred maximum-likelihood (ML) inferences were performed to take

phylogenetic uncertainty into account for subsequent ancestral state reconstruction analyses, using the Robinson–Foulds distance metric and the MRE-based bootstrap test[107] to test if 100 ML trees were sufficient (Supplementary Note 1). We opted to include only the long-read environmental sequences in our phylogenies because they better represent environmental diversity (compared to reference databases which are more biased towards culturable organisms and marine environments[108]) and because very few 18S–28S sequences can otherwise be ascertained to derive from the same organism. The final tree along with metadata was visualized using the anvi'o interface[109] and then modified in Adobe Illustrator v.24.2 to label clades.

**Short-read datasets.** *Datasets collected.* Short-read data corresponding to the V4 hypervariable region were retrieved from 22 publicly available metabarcoding datasets. Data were considered if the following criteria were fulfilled: (1) samples were collected from soils, freshwater or marine habitats; (2) there was clear association between samples and environment (no data from estuaries where salinity fluctuates); and (3) data publicly available or authors willing to share. The search for studies was not meant to be exhaustive and the datasets included in this work were identified and collected by the end of October 2020, unless specified otherwise. A list of these datasets can be found in Supplementary Table 3.

*Processing short-read data and clustering into OTUs.* Raw sequence files and metadata were downloaded from NCBI SRA web site (https://www.ncbi.nlm.nih.gov/Traces/study/) when available or obtained directly from the investigators. Information about the study and the samples (substrate, size fraction and so on), as well as the available metadata (geographic location, depth, date, temperature and so on), were stored in three distinct tables in a custom MySQL database. For each study, raw sequences files were processed independently de novo. Primer sequences were removed using Cutadapt[96] (maximum error rate = 10%). Amplicon processing was performed under the R software[110] using the dada2 package[90]. Read quality was visualized with the function plotQualityProfile. Reads were filtered using the function filterAndTrim, adapting parameters (truncLen, minLen, truncQ, maxEE) as a function of the overall sequence quality. Merging of the forward and reverse reads was done with the mergePairs function using the default parameters (minOverlap = 12, maxMismatch = 0). Chimaeras were removed using removeBimeraDenovo with default parameters. Taxonomic assignment of ASVs was performed using the assignTaxonomy function from dada2 against the PR$^2$ database[30] v.4.12 (https://pr2-database.org). ASV assignation and ASV abundance in each sample were stored in two tables in the MySQL database. ASV information was retrieved from the database using an R script. Data are available from the metapr2 database (https://shiny.metapr2.org/)[111].

ASVs from each environment (freshwater, soil, marine euphotic and marine aphotic) were clustered into OTUs at 97% similarity using VSEARCH[91], to make the size of the dataset more manageable for subsequent phylogenetic analyses. Identical or near-identical sequences can often be found in multiple habitats but these sequences do not necessarily represent generalists; instead some can be cases of very recent transition events (for example, refs. [51,52]) and we therefore chose to cluster the sequences of each environment individually to account for such cases. Identical sequences in multiple habitats can also be the result of contamination (for example, refs. [36,112]). Therefore, to be conservative in what was considered to be present in an environment, we retained only those OTUs that were composed of at least 100 reads or were present in at least two distinct samples.

*Testing for primer bias.* The 22 short-read datasets selected for this study were generated using nine different primer pairs (Supplementary Table 3). As no primer pair can amplify all taxa equally well, using multiple primer sets can bias the microbial communities obtained[113], thereby impacting downstream analyses on habitat transitions. To assess whether the different primer sets used lead to primer bias (certain taxa being detected in one habitat but not another), we tested each primer pair in an in silico analysis using the PR$^2$ primers database[113] (Supplementary Fig. 14). Firstly, nearly half of the datasets (10/22) were generated using one primer pair (primer set 8; TAReuk454FWD1 and TAReukREV3), spanning all four habitats. Secondly, most of the variation in primer sets comes from freshwater studies, which have used seven different primer sets in total. When not allowing any mismatches, different primer sets displayed reduced affinities for different eukaryotic clades (for example, primer set 16 amplifies <25% of rhizarian sequences in PR$^2$). However, no habitat has been surveyed by primers all biased against the same clade. Secondly, we note that PCR amplifications do not always correspond exactly to in silico analyses: some groups with mismatches against the primers can be amplified in the laboratory and vice versa. Therefore, we also performed analyses while allowing for four mismatches (corresponding to the default error-tolerance in Cutadapt[96] of 0.1 assuming all primers are 20 bp long). In this case, the primer pairs are largely equivalent, except for excavates (which we did not analyse in this study on account of high phylogenetic uncertainty). We therefore do not expect taxa to be missed in a certain habitat due to primer bias.

**Phylogenetic placement on global eukaryote phylogeny.** Short-read OTUs were aligned against the long-read alignment (section 'Maximum-likelihood analyses of the global eukaryotic dataset') using the phylogeny-aware alignment software

PaPaRa[114]. Misaligned sequences were systematically checked and removed. OTUs from the four environments were then phylogenetically placed on the global eukaryote tree (the tree with the highest likelihood) using EPA-ng[33]. OTUs with high EDPL (expected distance between placement locations) indicate uncertainty in placement and were filtered out with the gappa command edpl[102]. The resulting jplace files were visualized with iTOL[115].

**Inferring clade-specific phylogenies with short- and long-read data.** To investigate clade-specific transition rates across the salt barrier, we inferred phylogenies for major eukaryotic groups. We considered only those clades that contained sufficient data to more precisely infer transition rates: both non-marine and marine taxa were present and there were at least 50 taxa present. This excluded taxa such as radiolarians (which contain no non-marine taxa), rigifilids (which contain only non-marine taxa) and tubulineans (which are predominantly non-marine with an extremely small proportion of marine taxa). After preliminary analyses, we also excluded the clades discobans and discoseans due to large topological differences in the resulting trees.

We extracted all short-read OTUs from the remaining 13 clades using the gappa subcommand extract. Short-read OTUs taxonomically annotated as anything other than the respective clade were discarded (for instance, we discarded sequences labelled as amoebozoans that were phylogenetically placed in apicomplexa). For each clade, we pruned the corresponding subtree (and an outgroup) from the global phylogeny with the best likelihood score. For each clade, we then inferred 100 ML phylogenies with RAxML (GTRCAT model), using the long-read subtree as a backbone constraint. We estimated how robust these trees were by estimating SH-aLRT[98] (Shimodaira–Hasegawa approximate likelihood-ratio test) support values in IQ-TREE[116] v.1.6.3.

**Analyses of habitat-preference evolution.** *Unifrac analyses.* To estimate whether microbial communities from various habitats were phylogenetically distinct, we calculated unweighted UniFrac distance[117] as implemented in mothur, between (1) marine and non-marine habitats, (2) marine euphotic, marine aphotic, soil and freshwater and (3) each sample sequenced with PacBio. Distances were estimated along the best ML global eukaryotic phylogeny with 1,000 randomizations to test for statistical significance.

Similarly, we estimated pairwise Kantorovich–Rubinstein distance (earth mover's distance) between the four habitats (soil, freshwater, marine euphotic and marine aphotic) using the gappa subcommand krd with the short-read placement files (jplace files) as input (section 'Phylogenetic placement on global eukaryotic phylogeny').

*Model test on global eukaryotic phylogeny.* To investigate whether transition rates vary between major eukaryotic clades, we compared a homogeneous model ($q_{M-NM}$ and $q_{NM-M}$ remain constant throughout the global eukaryotic tree: a single rate regime) against a heterogeneous model ($q_{M-NM}$ and $q_{NM-M}$ estimated separately for each major eukaryotic clade; that is, multiple rate regimes) on the global eukaryotic phylogeny. These models were compared using Markov Chain Monte Carlo (MCMC) analyses in BayesTraits v.3.0.2 (refs. [118,119]) in a reversible-jump framework to avoid over-parameterization[120]. Briefly, under this framework, the Markov chain samples the posterior distribution of different models of evolution as well as the posterior distributions of the parameters of these models: no parameter restrictions were applied and the Markov chain simultaneously tested 'equal rates' and 'unequal rate' models, thereby integrating results over all possible model formulations weighted by their probabilities. Following the analysis in ref. [121], we used 50 stones and a chain length of 5,000 to obtain marginal likelihood for each model using stepping stone method[122] and a log Bayes factor (2 × difference of log marginal likelihoods) of ten or more was used to favour the heterogeneous model over the homogeneous model.

Before final analyses in BayesTraits, we tried several prior distributions for transition rates (using a hyperprior approach to reduce uncertainty about prior choice[120]). Specifically, we compared gamma hyperpriors with exponential hyperpriors using different values. While the different priors produced qualitatively similar results, we found the exponential hyperprior to be most suitable. All BayesTraits analyses were therefore carried out using an exponential hyperprior with the mean seeded from a uniform distribution between 0 and 2 in a reversible-jump MCMC framework, integrating results over all possible model formulations, weighted by their probabilities. All ancestral state reconstruction analyses were carried out on 100 inferred phylogenies to take phylogenetic uncertainty into account and were repeated thrice to check for convergence. Additionally, we assessed whether the variance of the transition rate parameters reached convergence as an increasing number of trees were sampled (Supplementary Fig. 15).

*Clade-specific transition rates.* We inferred clade-specific transition rates along the clade-specific phylogenies (long-read data + short-read data), on account of these being more complete. The metadata for each taxon was used to label it as either marine or non-marine. We ran 1 million generations on each tree (100 million generations in total) with 0.5 million generations discarded as burn-in, using the same hyperpriors as described above. For each clade, we also inferred the

global transition rate, regardless of the direction of transition. This was achieved by normalizing the QMatrix[123,124], with all other parameters unchanged. These analyses also allowed us to infer the ancestral state of each major eukaryotic clade.

*Sensitivity analyses for sampling efforts.* The short-read datasets included in this study were uneven in terms of number of samples and sequencing depth (Supplementary Table 4). As a result, the number of OTUs for soil and marine euphotic habitats were much higher compared to freshwater and marine aphotic environments. To test whether varying sampling efforts influences the clade-specific transition rates, we carried out a sensitivity analysis where we generated subsets of the short-read data by randomly removing 5%, 10%, 20%, 30%, 50% and 70% of (1) soil, (2) marine euphotic and (3) soil and marine euphotic OTUs (Supplementary Fig. 8). Five replicates for each subset were generated using SeqKit v.0.15.0 using the command sample. For each clade, we dropped tips corresponding to the OTUs removed from each subset using the custom script prune.py[88] and re-ran BayesTraits as before. In total, we estimated transition rates for 18 conditions×5 replicates×14 clades×100 trees per clade=126,000 trees.

*Inferring ancestral states of deep nodes and the last common ancestor of eukaryotes.* To infer the ancestral habitats at deeper nodes (including the origin of eukaryotes), we modelled habitat evolution along the global eukaryotic phylogeny using the better suited, heterogeneous model. Analyses were run for 500 million generations, with 5 million generations spent on each tree and 200 million generations were discarded as burn-in. Analyses were carried out after rooting the tree at Discoba and at Amorphea to take uncertainty about the root into account.

*Visualizing scenarios of habitat evolution.* Most ancestral state reconstruction programmes do not explicitly calculate the ancestral state at internal nodes (but integrate over all possibilities). To visualize habitat evolution, we used PastML, a maximum-likelihood ancestral state reconstruction programme which calculates the state at each internal node and also generates a concise visual summary of the clade[35]. For each major eukaryotic clade, we ran PastML on 100 trees. Visualizations for several trees were checked manually to assess if they displayed similar histories and one visualization was chosen randomly for display in Supplementary Fig. 7.

*Counting number and relative timing of transitions.* We converted all clade-specific phylogenies into relative chronograms (with the age of the root set to 1) using Path8 (ref. [37]) which is suitable for large phylogenies. We ran PastML on these phylogenies (as before) and used custom scripts[88] to count the number of marine/non-marine transitions. For each transition, we calculated the distance to the root to obtain relative timing of transition. We also validated our results with TreePL[125] v.1.0 (Supplementary Fig. 16). Results were largely similar to those obtained with Path8 but with transition events more shifted towards recent time.

**Network analyses.** To check that our results about transition rates and timings were not biased by phylogenetic inference from sequences with poor phylogenetic signal, we constructed sequence similarity networks. These networks were constructed using representative 18S sequences of the long-read OTUs. Briefly, we performed all-against-all BLAST searches and generated networks using a coverage threshold of 75 and sequence identity thresholds of 80, 85, 90, 95, 97. Networks were visualized on Cytoscape[126]. Assortativities were calculated using scripts available at https://github.com/MiguelMSandin/SSNetworks and then plotted in R using ggplot[127].

**Reporting summary.** Further information on research design is available in the Nature Research Reporting Summary linked to this article.

## Data availability

New sequence data generated for this study were deposited at ENA under the accession number PRJEB45931, while data from Sequel I (generated in ref. [28]) were deposited under the accession number PRJEB25197. The PR2-transitions database, annotated 18S and 28S OTU sequences, clustered short-read metabarcoding sequences used in this study and all trees have been deposited in an online repository[31]. Unclustered short reads are available from the metapr2 database (https://shiny.metapr2.org/)[111].

## Code availability

All custom code used in this study is available on Zenodo[88] with the identifier https://doi.org/10.5281/zenodo.6656264. Code for analysing sequence similarity networks is available at https://github.com/MiguelMSandin/SSNetworks.

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

## Acknowledgements

We thank A. Rosling, H. Urbina and M. Cafaro for kindly providing DNA from soil samples collected in Sweden and Puerto Rico. We thank the pilots of the deep-sea HOV 'Jiao Long Hao' and the crew of the R/Vs 'Xiang Yang Hong 09' for their professional service during cruise DY37II to collect samples from the Mariana Trench. We are grateful to the Swedish Infrastructure for Ecosystem Science (SITES) for collecting samples from Swedish lakes and Swedish Meteorological and Hydrological Institute for collecting a sample from the North Sea. Marine sampling was supported by the Spanish Ministry of Economy, Competitiveness projects Malaspina-2010 (CSD2008–00077) and ALLFLAGS (CTM2016-75083-R). We would like to thank E. Coulier for her help with optimizing the long-range PCRs. We thank O. V. Petterson and C. Tellgren-Roth for designing fusion primers for long-read amplification. We thank M. M. Sandin for his help with network analyses and advice on metabarcoding analysis and J. Boman for help with awk scripting. We thank J. del Campo for his advice on updating taxonomy for the custom PR2_transitions database. We thank the ABIMS platform of FR2424 (CNRS, Sorbonne Université) for bioinformatics resources. We would like to acknowledge support of the National Genomics Infrastructure (NGI)/Uppsala Genome Center and UPPMAX for providing assistance in massive parallel sequencing and computational infrastructure (SNIC 2021/5-302). Work performed at NGI/Uppsala Genome Center has been funded by RFI/VR and Science for Life Laboratory, Sweden. Finally, we are grateful to the Science for Life Laboratory for supporting this work in the laboratory of F.B.

## Author contributions

F.B. and M.J. conceived the project. F.B. supervised the project. M.J., H.J., S.P. and R.M. collected samples and extracted DNA. M.J. carried out long-range PCRs and processed the PacBio data. C.B. and D.V. collected and processed short-read metabarcoding data. A.O. performed comparisons of long- and short-read metabarcoding data. M.J. and C.B. performed phylogenetic and ancestral state reconstruction analyses. M.J and F.B. wrote the first draft of the manuscript and all authors read and commented on the manuscript.

## Funding

## Competing interests

The authors declare no competing interests.

## Additional information

**Extended data** is available for this paper at https://doi.org/10.1038/s41559-022-01838-4.

**Correspondence and requests for materials** should be addressed to Fabien Burki.

**a**

| | reads | | | ASVs | | | Total | |
|---|---|---|---|---|---|---|---|---|
| Dataset | 49/DCM | 49/Meso | 76/Surf | 49/DCM | 49/Meso | 76/Surf | reads | ASVs |
| mTags | 846 | 822 | 470 | - | - | - | 2138 | - |
| V4 | 77451 | 58953 | 147493 | 1740 | 1183 | 1800 | 283897 | 3753 |
| V9 | 46004 | 23642 | - | 2034 | 1127 | - | 69646 | 2777 |
| PacBio | 273073 | 642804 | 149790 | 4728 | 2277 | 1357 | 1065667 | 8362 |

**b**

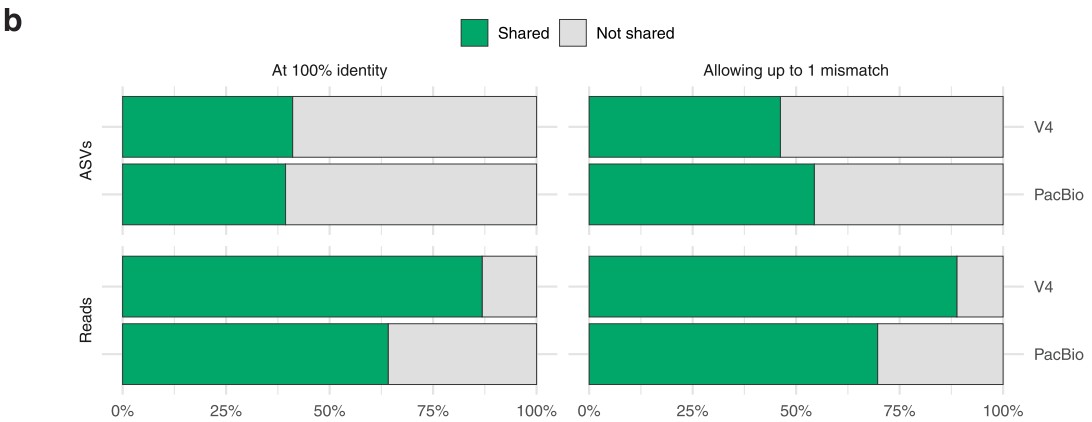

**Extended Data Fig. 1 | Shared ASVs between PacBio and Illumina sequencing.** Comparison of PacBio and Illumina sequencing. PacBio amplicons were compared with metagenomes (mTags), V4 amplicons, and V9 amplicons from three marine samples corresponding to the pico size fraction from the Malaspina expedition[27]. Station 76|Surface did not have V9 amplicon data. ASVs = Amplicon Sequence Variants. **(a)** Number of reads and ASVs for each sample for each marker. The mTags represent sequence length of ca. 100 bp, so no ASV level is available, as this short length does not give enough resolution. More PacBio sequences were generated for each sample compared to Illumina sequences. **(b)** Comparison of PacBio ASVs (that is de-noised, preclustered sequences) with the ones given by V4 amplicons. A similar comparison with V9 ASVs was not carried out as not all samples had V9 Illumina data available. Around half of the sequences were shared, which represented the majority of reads.

**a**

## Metagenomes versus other sequencing efforts
Relative abundances for each group separated by sample

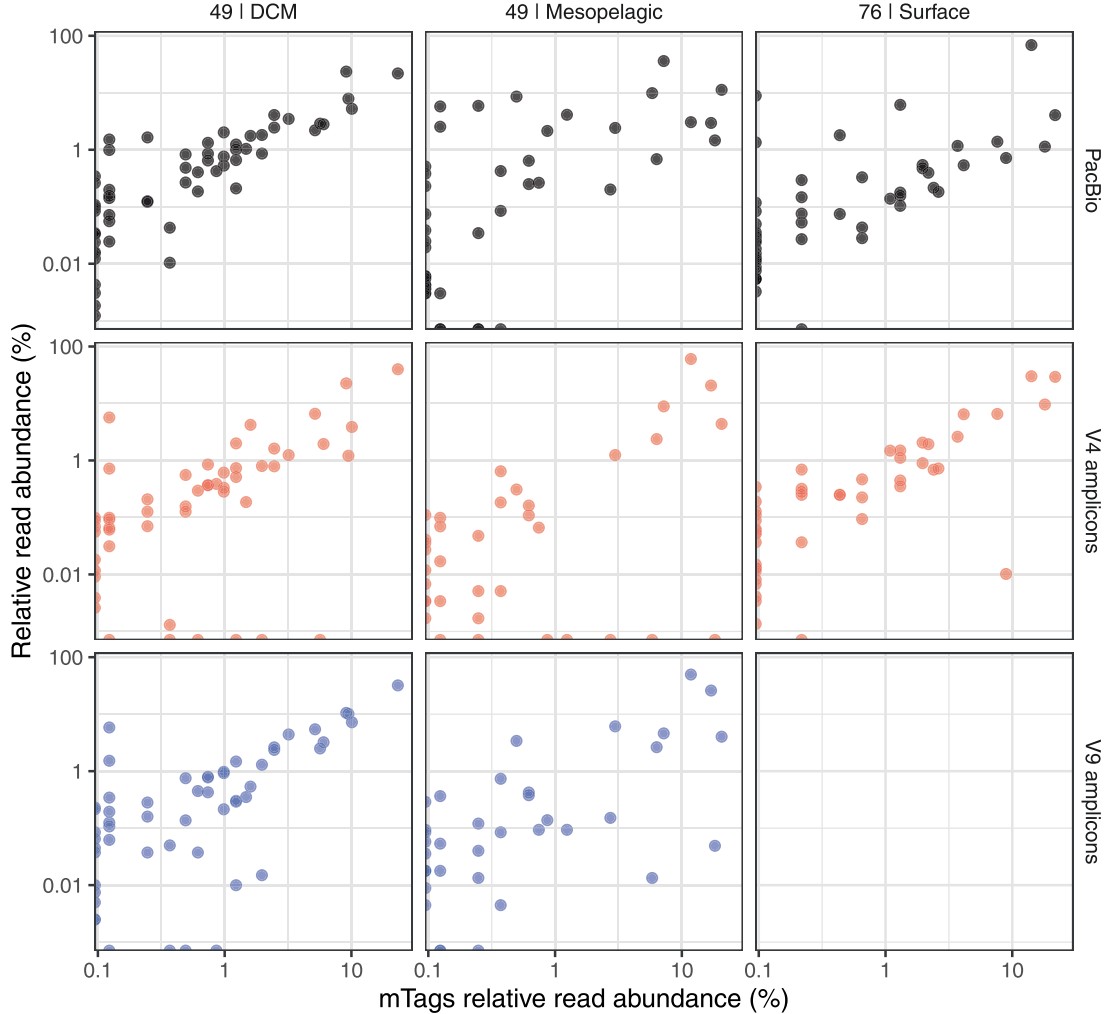

**b**

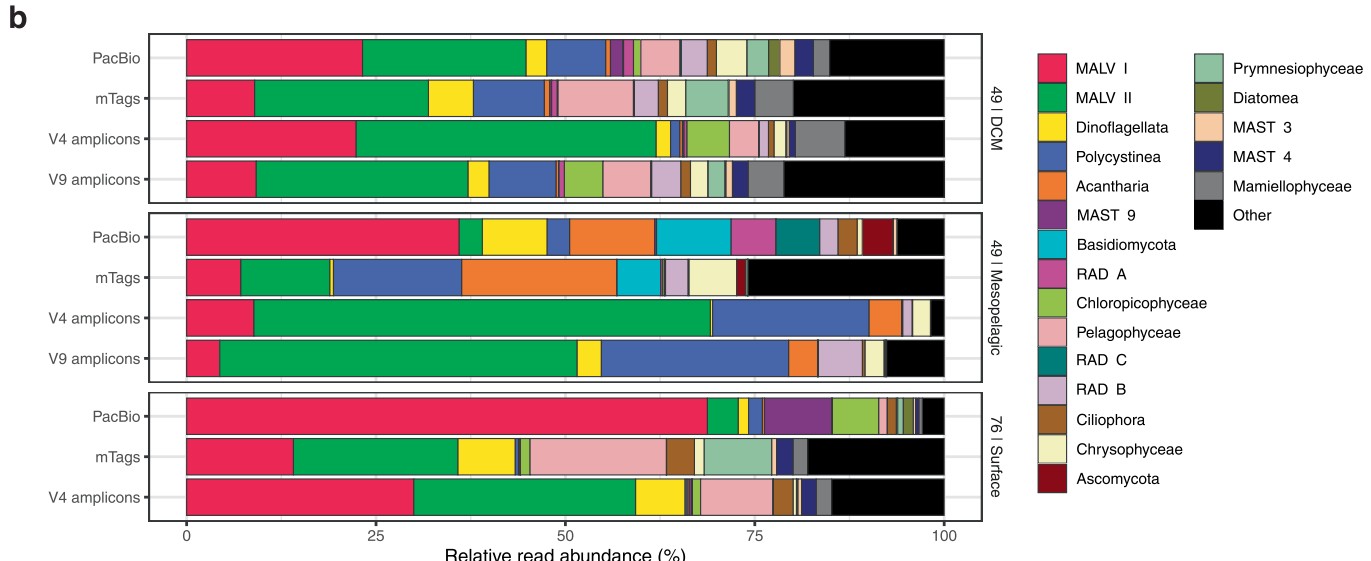

**Extended Data Fig. 2 | See next page for caption.**

**Extended Data Fig. 2 | Metagenomes versus other sequencing efforts.** Comparison of the eukaryotic communities retrieved by PacBio and Illumina sequencing (V4, V9, and 18 S reads retrieved from metagenomic data) of three marine samples (See Extended Data Fig. 1). **(a)** Comparison of mTags (which should represent a snapshot of the community unbiased by PCR) with the other datasets. Groups explaining the majority of reads are detected at comparable abundances. Points at the margins represent taxa that are found in one dataset but not in the other; the line of dots along the y-axis represent groups not present in mTags, but present in other datasets; and along the x axis we see groups that are present in mTags but not in the other datasets. For instance in the 49|DCM panels, there are some groups recovered by mTags that V4/V9 amplicons cannot detect (blue and red points at the bottom). Groups detected by PacBio in 49|DCM but missed by V9 include: MAST-25, MOCH-1, Marine-Opisthokonts; whereas groups missed by V4 include: kinetoplastids, discoseans, diplonemids, pyrmnesiophytes, Marine-Opisthokonts, and Basal-Fungi. Fewer black points (PacBio) at the bottom of the panels, indicates that PacBio is detecting groups that are missed by metabarcoding with V4/V9 sequencing. **(b)** Overall comparison of the relative abundances at the group level (excluding Charophyta, Metazoa and Nucleomorphs). The primer pair used for long-read sequencing seem to preferentially amplify MALV-I, but the overall community structure that PacBio is retrieving is reasonable with the other sequencing approaches.

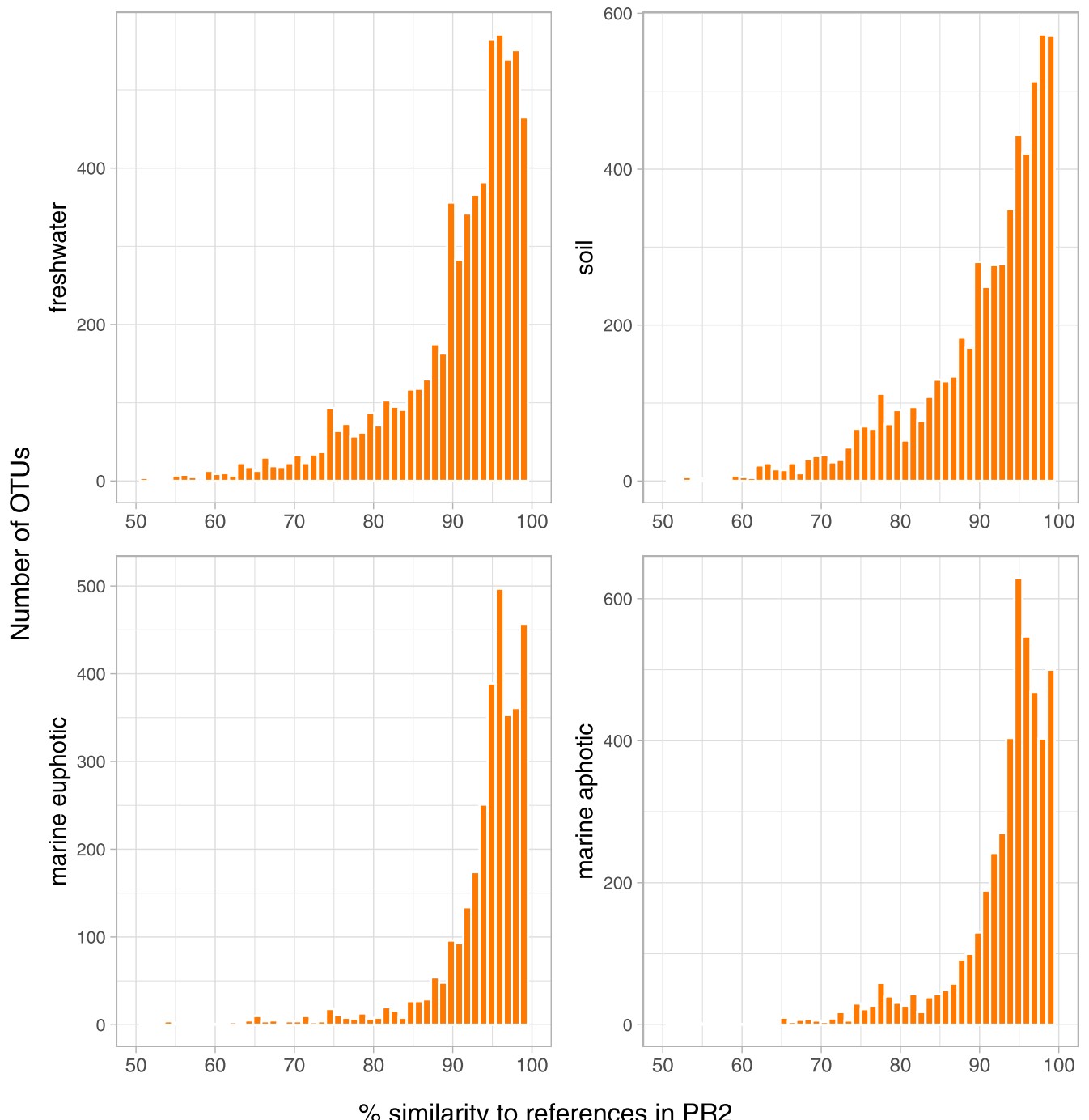

**Extended Data Fig. 3 | Percentage similarity of OTUs to references in PR².** Percentage similarity of OTUs (18 S sequence only) against reference sequences in the PR2 database[30], as determined by vsearch global search[91]. All sequences (OTU queries and references) were trimmed with primers 3NDF and 1510 R so that they spanned the same region.

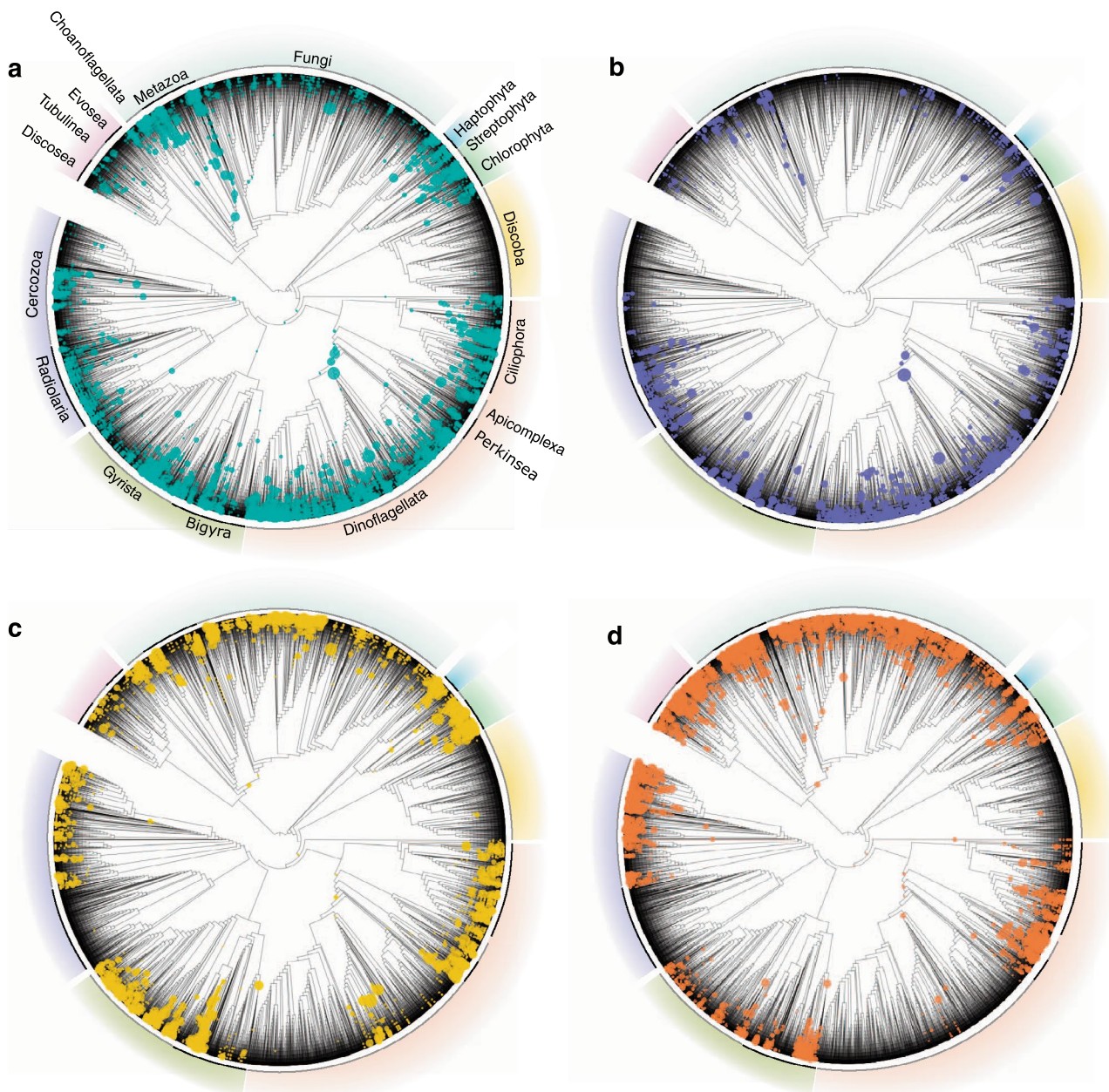

**Extended Data Fig. 4 | Phylogenetic placement of short-read OTUs on global eukaryotic phylogeny.** Phylogenetic placement of short-read OTUs onto the long-read, global eukaryotic reference phylogenetic tree (in Fig. 1). The upper two panels represent marine environments **(a**, marine euphotic; **b**, marine aphotic), while the lower two panels showcase non-marine placements (**c**, freshwater; **d**, soil). Visualization of the placement files was done through the interactive Tree of Life[115], and the size of each circle represents the number of placements on that particular branch weighted by the likelihood weight ratios.

**a**

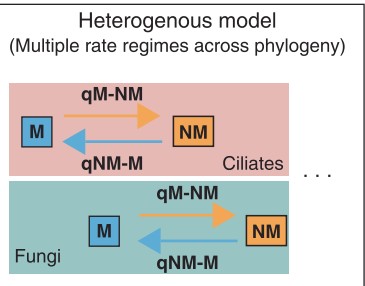

Homogenous model
(Single rate regime across phylogeny)

**qM-NM**

M → NM

**qNM-M**

qM-NM and qNM-M estimated over global phylogeny in Fig. 1

RevJump MCMC integrates over all possible models (qM-NM = qNM-M, qM-NM ≠ qNM-M, qNM-M = 0, qM-NM = 0), visiting each model in proportion to its posterior probability. (See Methods for detail).

Heterogenous model
(Multiple rate regimes across phylogeny)

**qM-NM**

M → NM    Ciliates

**qNM-M**

...

**qM-NM**

Fungi    M → NM

**qNM-M**

Different models of evolution fitted to different clades of the tree using RevJump MCMC.

**b**    Transition rates between habitats inferred from global eukaryotic phylogeny using homogenous model

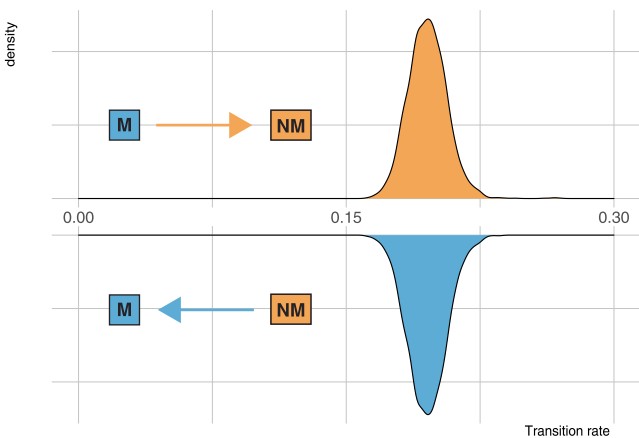

**c**    Comparison of log-likelihood of homogenous and heterogenous models

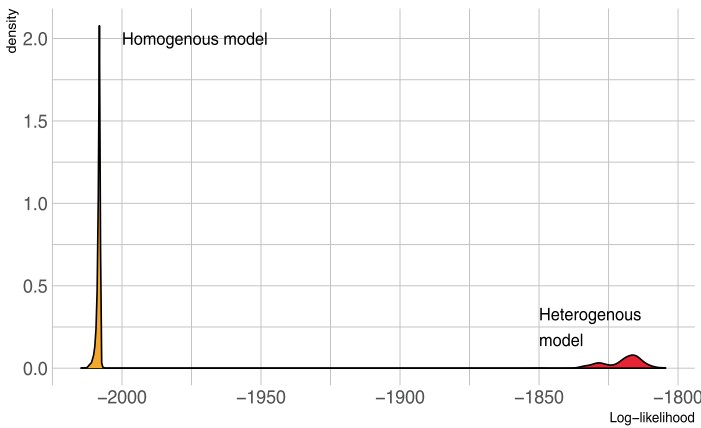

**Extended Data Fig. 5 | See next page for caption.**

**Extended Data Fig. 5 | Testing models of habitat transitions on global eukaryotic phylogeny. (a)** A graphical representation of the homogenous and heterogeneous models tested. The homogenous model involves a single rate regime over the tree (that is $q_{M-NM}$ and $q_{NM-M}$ have constant values). No restrictions are placed on the parameters; so $q_{M-NM}$ and $q_{NM-M}$ are allowed to be equal or unequal to each other. The heterogeneous model estimates a separate $q_{M-NM}$ and $q_{NM-M}$ for every major eukaryotic lineage (defined in this paper as rank 4 in the PR2-transitions database, for example Ciliates, Dinoflagellates, Fungi, etc.) that had at least 50 taxa and contained both marine and non-marine taxa. **(b)** Instantaneous transition rates from marine to non-marine habitats ($q_{M-NM}$) and vice versa ($q_{NM-M}$) when using a homogenous model over the global eukaryotic phylogeny (in Fig. 1). An equal rates model formulation ($q_{M-NM} = q_{NM-M}$) was found to have a higher posterior probability and was therefore sampled more frequently (100% of the time) by the reversible-jump Markov chain. **(c)** Comparison of the posterior probability of log-likelihoods when using the simple, homogenous model and the heterogeneous model. The plot shows that the heterogeneous model had a much better fit, indicating that rates of habitat evolution vary strongly across the eukaryotic tree of life.

**a**

**b**

**c**

**Extended Data Fig. 6 | Examples of transitions detected by the incorporation of short-read data.** Three examples of transitions detected by the incorporation of short-read data in our phylogenies that would otherwise have been missed. Shades of blue/purple represent marine sequences, while shades of orange/red represent non-marine taxa. **(a)** A clade of marine centrohelids is detected in purple (including sequences from the Malaspina expedition, Ocean Sampling Day, Tara Oceans, and Mariana Trench datasets). **(b)** A clade of marine chytrids is detected mainly from Ocean Sampling Day datasets. **(c)** A clade of non-marine haptophytes is detected mainly from the Swiss Soils and Neotropical soil datasets. Such cases were spread throughout the eukaryotic phylogeny.

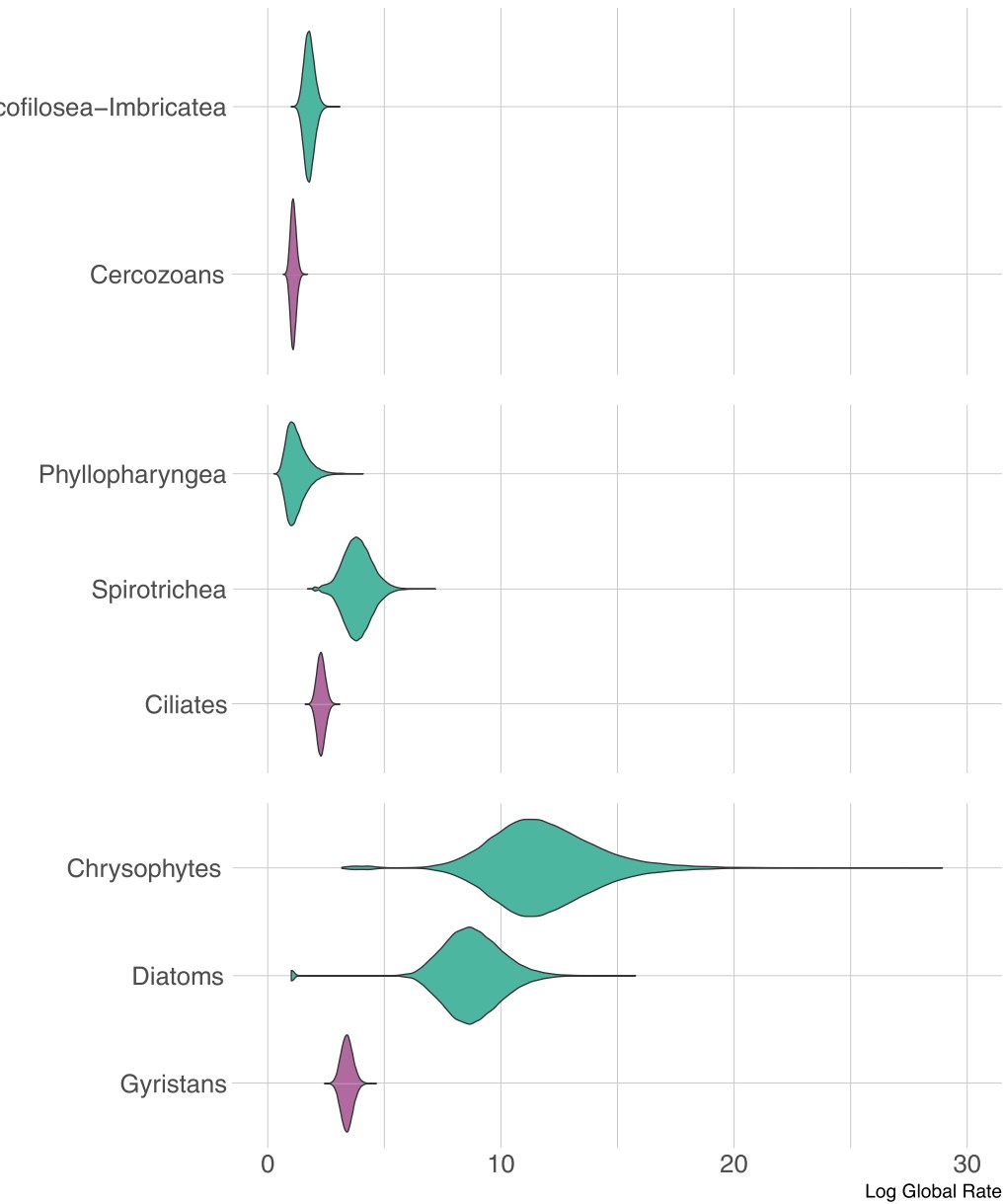

**Extended Data Fig. 7 | Heterogeneity of habitat transition rates within clades.** Posterior probability distributions of the global habitat evolution rates of selected eukaryotic classes. In purple, are the global transition rates of three major eukaryotic lineages (Cercozoans, Ciliates, Gyristans) as shown in Fig. 2, and in green, are the global transition rates for selected clades within these lineages. From this analysis, we can see that Thecofilosea+Imbricatea tend to transition across the salt barrier faster than Cercozoans on the whole. Spirotrich ciliates have higher transition rates than ciliates on average, and chrysophtes (golden algae) and diatoms seem to have the highest transition rates across protists.

| | |
|---|---|

# Reporting Summary

## Statistics

For all statistical analyses, confirm that the following items are present in the figure legend, table legend, main text, or Methods section.

| n/a | Confirmed | |
|---|---|---|
| ☐ | ☒ | The exact sample size (*n*) for each experimental group/condition, given as a discrete number and unit of measurement |
| ☐ | ☒ | A statement on whether measurements were taken from distinct samples or whether the same sample was measured repeatedly |
| ☐ | ☒ | The statistical test(s) used AND whether they are one- or two-sided *Only common tests should be described solely by name; describe more complex techniques in the Methods section.* |
| ☒ | ☐ | A description of all covariates tested |
| ☐ | ☒ | A description of any assumptions or corrections, such as tests of normality and adjustment for multiple comparisons |
| ☐ | ☒ | A full description of the statistical parameters including central tendency (e.g. means) or other basic estimates (e.g. regression coefficient) AND variation (e.g. standard deviation) or associated estimates of uncertainty (e.g. confidence intervals) |
| ☐ | ☒ | For null hypothesis testing, the test statistic (e.g. *F*, *t*, *r*) with confidence intervals, effect sizes, degrees of freedom and *P* value noted *Give P values as exact values whenever suitable.* |
| ☐ | ☒ | For Bayesian analysis, information on the choice of priors and Markov chain Monte Carlo settings |
| ☒ | ☐ | For hierarchical and complex designs, identification of the appropriate level for tests and full reporting of outcomes |
| ☒ | ☐ | Estimates of effect sizes (e.g. Cohen's *d*, Pearson's *r*), indicating how they were calculated |

*Our web collection on statistics for biologists contains articles on many of the points above.*

## Software and code

Policy information about availability of computer code

| Data collection | No software was used for data collection. |
|---|---|
| Data analysis | SMRT Link v8.0.0.79519, mothur v1.39.5, DADA2 v1.14.1, VSEARCH v2.3.4, Blast v2.9.0, Barrnap v0.9, RAxML v8.2.10, RAxML-NG v0.9.0, EPA-ng v0.3.5, gappa v0.6.0, MAFFT v7.310, trimAl 1.2rev59, TreeShrink v1.3.5, cutadapt v3.3, PaPaRa v2.5, BayesTraits v3.0.2, pastml 1.9.30, PATHd8, Cytoscape v3.8.2. All custom scripts used in this study are available at https://doi.org/10.5281/zenodo.6656264 under a MIT license. |

For manuscripts utilizing custom algorithms or software that are central to the research but not yet described in published literature, software must be made available to editors and reviewers. We strongly encourage code deposition in a community repository (e.g. GitHub). See the Nature Portfolio guidelines for submitting code & software for further information.

## Data

Policy information about availability of data

All manuscripts must include a data availability statement. This statement should provide the following information, where applicable:
- Accession codes, unique identifiers, or web links for publicly available datasets
- A description of any restrictions on data availability
- For clinical datasets or third party data, please ensure that the statement adheres to our policy

New sequence data generated for this study were deposited at ENA under the accession number PRJEB45931, while data from Sequel I (generated in reference 28) were deposited under the accession number PRJEB25197. The PR2-transitions database, annotated 18S and 28S OTU sequences, clustered short read metabarcoding sequences used in this study, and all trees have been deposited in a Figshare repository (https://doi.org/10.6084/m9.figshare.15164772.v3). All custom code is available in a Zenodo repository (https://doi.org/10.5281/zenodo.6656264).

# Field-specific reporting

Please select the one below that is the best fit for your research. If you are not sure, read the appropriate sections before making your selection.

☐ Life sciences ☐ Behavioural & social sciences ☒ Ecological, evolutionary & environmental sciences

For a reference copy of the document with all sections, see nature.com/documents/nr-reporting-summary-flat.pdf

# Ecological, evolutionary & environmental sciences study design

All studies must disclose on these points even when the disclosure is negative.

| Study description | A phylogenetic analysis to investigate habitat preference evolution across the tree of eukaryotes using long-read and short-read environmental sequencing data. |
|---|---|
| Research sample | Long-read metabarcoding data were generated from 21 samples covering soils, freshwater, marine euphotic, and marine aphotic habitats (details provided in Supplementary Table 1). With these data, we aimed to get a representative view of microbial eukaryotic diversity in each habitat. Existing short-read metabarcoding data were obtained from 22 publicly available datasets (details provided in Supplementary Table 3). |
| Sampling strategy | We ensured that the 21 samples sequenced in this study were not skewed towards a particular habitat, and used general eukaryotic primers were amplify the ribosomal DNA operon in order to capture the broad eukaryotic community. We then increased the covered eukaryotic diversity by using data from 22 publicly available short-read metabarcoding datasets spanning marine and non-marine habitats. |
| Data collection | Data were generated as part of this study, or downloaded from public databases. |
| Timing and spatial scale | Five freshwater samples (four planktonic and one sediment sample) were obtained from lakes in Sweden and permafrost thaw ponds in Canada in 2019 and 2014 respectively. Soil samples were sourced from Sweden (2019 and 2013), Puerto Rico (2013), UK (2015 and 2008), and Tibet (2011). Marine samples were obtained from the Mariana Trench (2016), the West Coast of Sweden (2019) and two stations from the Malaspina expedition (several depths; 2011). Sample coordinates and sampling dates are provided in Supplementary Table 1. It should be noted that sampling procedures for samples collected outside Sweden have been described in previous publications (see Supplementary Table 1 for references). |
| Data exclusions | We excluded 18S-28S sequences likely representing chimeras, artefacts from all samples and potential contaminants due to barcode mixing in one marine sample (Ms2-DCM), by manually inspecting 18S and 18S-28S phylogenies. To be conservative in what we considered to be "present" in a habitat, we excluded short-read OTUs that had low abundance (<100) or were only present in one environmental sample. |
| Reproducibility | All analyses are described in the Methods section with software versions and parameters used. Curated, labeled, 18S-28S OTU sequences are provided on FigShare, as are clustered short-read OTU sequences. |
| Randomization | We performed 1000 randomizations in the UniFrac analyses to test whether microbial communities from each habitat were phylogenetically distinct. |
| Blinding | No blinding was applied as no statistical tests were performed where blinding could be applied. |

Did the study involve field work? ☒ Yes ☐ No

## Field work, collection and transport

| Field conditions | Marine surface waters, deep marine waters, soils, and freshwater lakes. Sampling dates and brief description of each sample is provided in Supplementary Table 1. |
|---|---|
| Location | Coordinates provided in Supplementary Table 1. |
| Access & import/export | Samples collected from outside Sweden have previously been described in previous publications (see Supplementary Table 1 for references); therefore no collecting permit was required for this study. |
| Disturbance | No disturbance was caused; only a small amount of water or soils were collected. |

# Reporting for specific materials, systems and methods

We require information from authors about some types of materials, experimental systems and methods used in many studies. Here, indicate whether each material, system or method listed is relevant to your study. If you are not sure if a list item applies to your research, read the appropriate section before selecting a response.

## Materials & experimental systems

| n/a | Involved in the study |
|-----|----------------------|
| ☒ ☐ | Antibodies |
| ☒ ☐ | Eukaryotic cell lines |
| ☒ ☐ | Palaeontology and archaeology |
| ☒ ☐ | Animals and other organisms |
| ☒ ☐ | Human research participants |
| ☒ ☐ | Clinical data |
| ☒ ☐ | Dual use research of concern |

## Methods

| n/a | Involved in the study |
|-----|----------------------|
| ☒ ☐ | ChIP-seq |
| ☒ ☐ | Flow cytometry |
| ☒ ☐ | MRI-based neuroimaging |

