## [Peer Review File · Nature Ecology & Evolution]

Peer Review Information

Journal: Nature Ecology & Evolution

Manuscript Title: Global patterns and rates of habitat transitions across the eukaryotic tree of life

Corresponding author name(s): Fabien Burki

Editorial Notes:

Reviewer Comments & Decisions:

Decision Letter, initial version:
--

17th January 2022

Dear Dr Burki,

Your manuscript entitled "Global patterns and rates of habitat transitions across the eukaryotic tree of life" has now been seen by 4 reviewers, whose comments are attached. The reviewers have raised a number of concerns which will need to be addressed before we can offer publication in Nature Ecology & Evolution. We will therefore need to see your responses to the criticisms raised and to some editorial concerns, along with a revised manuscript, before we can reach a final decision regarding publication.

We therefore invite you to revise your manuscript taking into account all reviewer and editor comments. Please highlight all changes in the manuscript text file.

* Include a "Response to reviewers" document detailing, point-by-point, how you addressed each reviewer comment. If no action was taken to address a point, you must provide a compelling

argument. This response will be sent back to the reviewers along with the revised manuscript.

* If you have not done so already please begin to revise your manuscript so that it conforms to our Article format instructions at <http://www.nature.com/natecolevol/info/final-submission>. Refer also to any guidelines provided in this letter.

[REDACTED]

Nature Ecology & Evolution is committed to improving transparency in authorship. As part of our efforts in this direction, we are now requesting that all authors identified as 'corresponding author' on published papers create and link their Open Researcher and Contributor Identifier (ORCID) with their account on the Manuscript Tracking System (MTS), prior to acceptance. ORCID helps the scientific community achieve unambiguous attribution of all scholarly contributions. You can create and link your ORCID from the home page of the MTS by clicking on 'Modify my Springer Nature account'. For more information please visit www.springernature.com/orcid.

[REDACTED]

Reviewers' comments:

Reviewer #1 (Remarks to the Author):

2Jamy et al examined global rates of transition between marine, terrestrial and freshwater habitats across eukaryotes, using long-read environmental sequencing. This study is a pioneer of its kind and opens very exciting avenues for understanding evolutionary transitions in eukaryotes globally. I think the ms is well-written, the analyses are well-designed and appropriately performed. I have a few suggestions that the Authors might want to consider for improving their ms.

The authors basically dismiss the first comparative analysis (lines 149-162) as 'hiding important variations'. I think these analyses should be reported better - even if they do provide give clear-cut evidence, more detailed reporting would make this section easier for readers to assess. For example, rate parameters inferred by BayesTraits could highlight whether there is an asymmetry in transition rates (see below).

Testing whether the 100 ML topologies are sufficient to cover topological uncertainty would be interesting. For example, do the authors get a plateau if they plot the number of bipartitions seen as a function of the number of trees?

Though I'm sure the Authors have done a good job in documenting diversity at the habitats they sampled for long read sequencing, there may still be groups undersampled in their phylogeny. In the discussion, it would be interesting to mention if undersampling exists in the dataset and how, if at all this might have influenced rate estimations and ancestral state reconstructions.

I'm a bit skeptical with the high rate of transitions inferred for fungi. As the Authors also discuss briefly (l219), fungal spores can reside in environments in which the fungus is not able to live, including marine habitats. The Authors' analyses of the number of transition events correctly take this into account, but those of transition rates do not. I still think there might be an overestimation in fungi, given the ubiquitous occurrence of their spores. I don't know if a sensitivity analysis or merely interpreting these results with a grain of salt would solve this issue, but I think there is room for reconsideration here.

Smaller comments:

l51 - strange phrase, maybe 'increasing <number of?> inferences?

l159 - at this point it would be interesting to report rate parameters from a model in which q_{MT} and q_{TM} can take independent values (but uniform across the tree). Is there a rate asymmetry?

l161-162 - please report rate values from these analyses. Or, if the global tree proved for some reason unsuitable for analyzing rates, please detail why. Is there too much masking of rate values due to opposite patterns between clades? Is there an overall pattern emerging?

l167 - how robust were those trees?

l170-176 - a parsimony based introduction of the frequency of transitions would be much easier to interpret. Because bayestraits rates do not have clear parsimony interpretation, I think reporting the number of transitions inferred by parsimony, in each clade, would make interpretation easier.

l201 - please indicate how significance was defined.

l273 - 'rhizarians'

fig4b - a marking is missing for the polytomic mrca of ichtyosporeans and metazoans - on purpose?

l308 - well, I'm not sure this ms showed that the salt barrier per se shaped euk diversity. It showed that euk diversification involved several transitions across this barrier.

3

I336 - I'm a bit skeptical with the fungal results, see above.

Reviewer #2 (Remarks to the Author):

Jamy and co-authors present here a global study that assess the transition (marine and terrestrial) pattern and rate across the eukaryotic tree of life. To cover the general diversity of marine (euphotic and aphotic) and terrestrial (freshwater and soils) ecosystem, they have used a metabarcoding approach combining long-read (4500bp, 18S, ITS and 28S, PacBio) and short-read (450bp, V4 regions of the 18S rRNA gene, Illumina, 454, 22 published dataset). They have built a phylogenetic reconstruction tree based on the twenty-one samples that were sequenced with PacBio. Then they have included the short-reads in the phylogenetic reconstruction by using an EPA-ng placement. They have used multiple statistical approaches (Unifrac, Model test, ancestral state reconstruction) to investigate the pattern and the variation of transition rates among the ecosystem between major eukaryotic clades. The keys finding of the study are 1) transition occurred in both directions in almost all major eukaryotic lineages 2) ancestral habitat reconstruction analyses suggest that the eucaryotes have first evolve in terrestrial environment. Overall, the study provides the first global evaluation of the role of the salt barrier in the evolution of the eucaryotes.

The manuscript is well written, The authors used the up to date tools of the field of research (Jamy et al., 2020) for the 18+3 samples (long-reads analyses) but also include 22 published dataset of short reads to complete the evaluation of the diversity. The description of the statistics, the rationale as well as the development of the manuscript is clear. The authors provide all the information (Raw Data, scripts, supplementary material, and references) to correctly redo the experiment and statistic. Nevertheless, I have highlighted some issues/limitations of the study that the authors must investigate and provide the necessary justifications before a potential acceptance in Nature Ecology and Evolution.

L371-408: Environmental samples for long-read metabarcoding and total DNA extraction

The authors used different approaches and methods to collect and analyze their samples. I understand the purpose of the study to remain at a global level, nevertheless each modification in the methodological procedure will induce potential biases in the interpretation of the results.

Different DNA extraction kit were used according to the sampling location. For example, for soils samples in rainforest soils (L383) the DNA were extracted using the Xpedition™ Soil/Fecal DNA miniprep and they have used the DNeasy PowerSoil Kit (Qiagen) for the samples from the Lake Erken. The choice of the DNA extraction kit can have an impact on the community profile that you will obtain (Santos et al., 2017).

Line 387-395: The authors combine plankton and sediment samples under the term "freshwater". Nevertheless, the community composition between this two environments can be really different (Forster et al., 2016). I suppose that this choice was made to cover more diversity of freshwater ecosystem? Moreover, the authors choose to combine freshwater and soils ecosystems and coin it under the term "terrestrial." Many authors (Singer et al., 2021; Bižić M. et al.) keep this two

4ecosystems separated. The aim of the article is to study the effect of the saline barrier, so instead of "terrestrial" I would prefer to use the term "no marine." This is way you can combine freshwater and soils.

Line 395: The quantity of water filtered, and the size of the mesh used to obtain the raw material are different between the different studies/location. This will also have an impact on the community profile obtain in the end.

All these biases will potentially impact the vision of the diversity. I suggest including a short paragraph to highlight, discuss and justify briefly all these elements inherent to every meta-analyses.

L500-555 Short read datasets

I noticed two main elements that will potentially influence the results and the conclusion of the study in the short read dataset processing. First the dataset of the short reads is completely unbalanced among the four studied ecosystem. In Sup Table 4, we can observe that there are about ten time more reads for the soil environment than for freshwater. I suppose that this fact is due to the limited numbers of study for freshwater environment. The same observation can be done for the marine euphotic and marine aphotic dataset. I recommend presenting basic analyses to provide the evidence that the numbers of dataset/sites/samples per ecosystems used in the study are enough in term of sequencing depth to cover the whole diversity of each of these environments.

The diversity obtain in the twenty-two datasets were obtain by using different couple of primers. This difference will preferentially amplify certain taxon as we can see in sup. Fig 7. As Dr. Daniel Vaultot (co-author of the article) have said in his last paper published in Molecular Ecology Resources: To cover a relevant and representative fraction of the protist community in a given study system, an informed primer choice is necessary, as no primer pair can target all protists equally well (Vaultot et al., 2021). The used of primers pair in the study are also unbalanced (Sup Table 3). With primers that were only present in a certain kind of ecosystems. This will lead to potentially the amplification of specific clades that remain undetected in the other environment. I would like to know if some transitions found in the study are not "just" due to a primers pair bias. The authors must provide the evidence that their results have not been impacted by this potential bias. And potentially redo their analyses by selecting only the taxa that are amplified by all the primers pairs.

Minor comments:

The point "3. Global 18S-28S phylogeny" in the github depository is a confusing. I do not understand why do you have "Amorphea.list" file for example? I suggest adding more explanation as sentences like: "inspected all alignments visually by eye," make the process difficult to duplicate.

The computation of the percentage of identity (Fig 1, Sup Fig 3) against the reference database (PR2) is not well explain in the methods section. The long reads contain the full 18S rRNA gene, but most of the reference sequences in PR2, are truncated according to the specific primers used to target the organisms. How did the authors take this fact in to account?

References: genus and species must be in "italic" (L.841: *Helkesimastix marina*)

PR2: the "2" of PR2 must be written as a superscript

Line 37: The term "Physicochemical" is related to the physical chemistry field of research. I suppose that the meaning of the sentence is more related to the physical (temperature, light irradiation...) and the chemical (nutrient content, pH...) properties of the environment? The term "Physicochemical" is often miss used in ecological studies, I recommend separating the terms as it is related to two different concepts.

Line 531: Specify the minimum numbers of samples where the OTU must be present.

Fig 1: To better visualize the range of the percentage of identity, I suggest adding a line that indicate 100% of similarity

Fig 1 and Sup Fig 7) I recommend using the same color codes and names between the figures to allow a better comparison between the figures.

Sup Fig 4) "Freshwater=Yellow"

Sup Fig 5) "Font size issue"

Sup table 3 and 4) Add the numbers of samples/sites

Bižić M., Klintzsch T., Ionescu D., Hindiyeh M. Y., Günthel M., Muro-Pastor A. M., et al. Aquatic and terrestrial cyanobacteria produce methane. *Science Advances* 6: eaax5343.

Forster, D., Dunthorn, M., Mahé, F., Dolan, J.R., Audic, S., Bass, D., et al. (2016) Benthic protists: the under-charted majority. *FEMS Microbiology Ecology* 92: fiw120.

Jamy, M., Foster, R., Barbera, P., Czech, L., Kozlov, A., Stamatakis, A., et al. (2020) Long-read metabarcoding of the eukaryotic rDNA operon to phylogenetically and taxonomically resolve environmental diversity. *Molecular Ecology Resources* 20: 429–443.

Santos, S.S., Nunes, I., Nielsen, T.K., Jacquiod, S., Hansen, L.H., and Winding, A. (2017) Soil DNA Extraction Procedure Influences Protist 18S rRNA Gene Community Profiling Outcome. *Protist* 168: 283–293.

Singer, D., Seppely, C.V.W., Lentendu, G., Dunthorn, M., Bass, D., Belbahri, L., et al. (2021) Protist taxonomic and functional diversity in soil, freshwater and marine ecosystems. *Environment International* 146: 106262.

Vaulot, D., Geisen, S., Mahé, F., and Bass, D. (2021) pr2-primers: An 18S rRNA primer database for protists. *Molecular Ecology Resources* n/a:

Dr. David Singer

Reviewer #3 (Remarks to the Author):

6The paper by Jamy et al. investigates habitat evolutionary transitions across the eukaryotic tree of life using a novel approach that combines long and short read metabarcoding, showing that eukaryotes might have evolved in non-saline habitats, the crossing of the salt barrier might promote diversification, and that such barrier has been crossed several hundred times. The methods overall are sound and innovative, and the conclusions seem appropriate based on the results.

I have a few comments outlined below:

The use of global for the long read data, for instance line 137, does not seem appropriate as the samples were obtained from only a handful of sites, even if they covered all major ecosystems. It is only justifiable to use the term global phylogeny when including the short read data as this really represents a global sample. Then it also becomes confusing when the authors refer to the global eukaryotic phylogeny. Do they mean the one including only long read data or the one they have added the short read data? My suggestion is to refer to it as the "long read data phylogeny" and avoid the use of global phylogeny, unless they show a large phylogeny with long + short read data, but it seems that the phylogenies including long + short read data are subtrees from the long read data phylogeny. An alternative would be to clearly specify what the term global means in global phylogeny.

Where there any taxa that can be found in more than one environment? If so, I wonder how this is reflected in the analyses? When doing this kind of analyses, I often find that the character coding for some taxa might not be very straightforward, and sometimes one taxon can be coded as both states, so for example in this particular study I wonder if there were OTUs found in more than one environment and how that was dealt with.

While PATHd8 is one of the few methods that allows dating big phylogenies, it is also known that other methods outperform it (e.g., r8s and TreePL). PATHd8 has been shown to underestimate divergence times, and to not be very reliable (Smith and O'Meara, 2012. *Bioinformatics*, Tao et al. 2020, *Efficient Methods for Dating Evolutionary Divergences in The Molecular Evolutionary Clock*). I suggest to corroborate the results with TreePL (which also works for big phylogenies, see Sun et al. 2020. *Nature communications* 11, 3333) to ensure that the results, particularly the transition rates are not biased due to the estimation time method.

Reviewer #4 (Remarks to the Author):

This manuscript presents a global phylogenetic analysis of salinity preference in eukaryotes. The authors construct a new eukaryotic phylogeny using long-read 18S-28S sequences, providing greater resolution and higher coverage than previous efforts, and then map habitats onto this tree. They find that in general, most lineages tend to be either freshwater or marine, consistent with infrequent ancient transitions, but at the same time they document many instances of more recent transitions. I find the most interesting result to be the clade-specific relative rates of transitions across different eukaryotic groups, and in particular the high rate in Fungi compared to other groups. This work will hopefully inspire future investigations of the biological traits and mechanisms underlying this variation

7across groups.

In general, I found this paper to be fun to read, interesting, and well written with clear figures, and in my opinion it will be of broad interest to protist biologists, evolutionary and paleobiologists, and ecologists. The new long-read dataset should be quite useful to other researchers. I think the methods are careful and generally well described, but I would like to see some mention of how sensitive the findings are to particular methodological choices. Below I offer some suggestions for clarity and for tempering some claims a bit.

1. I am a bit skeptical of the ancestral state reconstruction going all the way back to the LECA. I would like to see these claims perhaps tempered a bit and/or a discussion of how robust these findings are to the inclusion of specific taxa, to specific topologies, and to specific model formulations.
2. One might argue that the choice of environments/samples here is far from "comprehensive" (line 73) and does not capture "all major ecosystems". I'm thinking of anoxic environments, animal-associated, deep sediments and subsurface, extremes of temp/pH/etc. How much of the known phylogenetic diversity is **not** captured by the datasets here? And does that influence the results?
3. Line 83-85: the PacBio data produces a similar profile for the DCM sample, but arguably not so much for the SRF and deep sample (the dot plots are log scale; the bars below don't look very similar). Perhaps discuss. Also I don't understand the bar plots in Fig S1b – would a Venn diagram showing the overlap be easier to read?
4. I don't entirely understand the reason for assigning taxonomy to each of the 18S-23S sequences before building the new tree. Is this related to the monophyly constraints? Or is it necessary because there are few (any?) reference sequences as landmarks in the new phylogeny?
5. By collapsing sequences into 97% OTUs, the methods are probably underestimating recent transitions. Might be worth highlighting in the discussion.
6. Discussion of timing of transition events (line 319+): it seems misleading to suggest that transitions are more frequent in recent time. Aren't there more opportunities as you go forward in time (given the extant taxa we have)?
7. I think the Discussion could touch upon functional traits that might influence the phylogenetic patterns described here. How might parasites/symbionts vs free living taxa differ in propensity for transitions? Photosynthetic vs heterotrophic vs mixotrophic? Nutrition, motility, dispersal ability, animal hosts, etc. Are there any examples from specific lineages that might be informative here?

Minor comments

1. Abstract last sentence: the paper doesn't directly address this. Consider modifying this sentence.
2. "environmental phylogeny" – 1st subheading in Results and Abstract line 25 – this phrase doesn't make sense. Do you mean a phylogeny based on environmental (as opposed to cultivars or type species) sequences?
3. Line 176 – extra "and"
4. Methods: please clarify the pooling of samples earlier on (c.f. line 419-420). Were pico/nano size fractions also pooled?

*****END*****

Author Rebuttal to Initial comments

Reviewer 1 (Remarks to the Author):

Jamy et al examined global rates of transition between marine, terrestrial and freshwater habitats across eukaryotes, using long-read environmental sequencing. This study is a pioneer of its kind and opens very exciting avenues for understanding evolutionary transitions in eukaryotes globally. I think the ms is well-written, the analyses are well-designed and appropriately performed. I have a few suggestions that the Authors might want to consider for improving their ms.

–Response: We thank the reviewer for their encouraging remarks and have made the changes recommended below.

The authors basically dismiss the first comparative analysis (lines149-162) as ‘hiding important variations’. I think these analyses should be reported better - even if they do provide give clear-cut evidence, more detailed reporting would make this section easier for readers to assess. For example, rate parameters inferred by BayesTraits could highlight whether there is an asymmetry in transition rates (see below).

–Response: Thank you for this comment. Although we reported (in the same paragraph, lines 158-159 in the first submission) whether the transition rates were symmetrical or asymmetrical, we agree that the results were not clearly presented, and we apologise for the confusion. Therefore, we have updated the corresponding supplementary figure (Supplementary Figure 10) with a new panel (Panel A) to clarify the homogenous and heterogenous models tested. We have also re-written this section as follows (lines 153-168):

“The above results confirm that the salt barrier leads to phylogenetically distinct eukaryotic communities. We next asked (i) how often have transitions between marine and non-marine habitats occurred during evolution, (ii) which eukaryotic lineages have crossed this barrier more frequently, and (iii) in which direction? To answer these questions, we calculated habitat transition rates across the global eukaryotic phylogeny by performing Bayesian ancestral state reconstructions using continuous-time Markov models³⁵. We first tested a homogenous model, where a single pattern of transition rates from marine to non-marine habitats (and vice versa) was estimated across all eukaryotes. The homogenous model returned a posterior density of log-likelihoods with a mean of -2008.45, and transitions from marine to non-marine habitats were found to be just as likely as the opposite direction across the tree $q_{M-NM} = q_{NM-M} = 0.19$ transitions per substitution per site; Supplementary Figure 10). However, the assumption of the homogenous model of uniform transition rates across the tree may be violated if there are large variations in habitat transition rates between groups. Indeed, a heterogeneous model where we estimated and separately for each major eukaryotic clade, presented a much better fit (log-likelihood score of -1819.91; Log Bayes Factor = 269.3; Supplementary Figure 10), indicating that habitat transition rates vary strongly across the tree.”

Testing whether the 100 ML topologies are sufficient to cover topological uncertainty would be interesting. For example, do the authors get a plateau if they plot the number of bipartitions seen as a function of the number of trees?

–Response: We thank the reviewer for this excellent suggestion. We performed the three analyses below to address this concern, which all indicate that 100 ML trees sufficiently represent the plausible set of topologies. These new results are now described in a new Supplementary Note 1, and a new Supplementary Figure 22, both referenced in the main text in lines 94, 557, and 680.

- i. We calculated relative pairwise Robinson-Foulds (RF) distances in raxml-ng (using the command --rf-dist), where an RF-distance of 0 indicates identical topologies, and 1 indicates that no bipartitions are shared. As we had only constrained 60 internal branches out of the possible 16818 branches, the upper bound on the possible pairwise RF-distance of two trees (T1 and T2) was close to 1 (calculation shown below).

10Maximum RF distance = $\frac{\text{total number of possible unshared bipartitions in } T1 \text{ and } T2}{\text{total number of bipartitions in both trees}}$

$$= \frac{(16818-60)(2)}{(16818)(2)} = 0.996$$

Our calculations showed that the 100 ML trees represented 100 unique topologies with an average relative RF distance of 0.0675, indicating that the trees are topologically very similar on average.

- ii. We performed the auto MRE-based bootstrapping test in raxml-ng using a cutoff value of 0.05 (-bsconverge --bs-cutoff 0.05). The algorithm (described in Pattengale et al. 2010¹) is typically employed to assess how many bootstrap replicates are enough by considering the variance in the tree set. Briefly, the tree set is split into two sub-sets with 1000 permutations, and the weighted RF distance is calculated between the majority-rule consensus trees of the two sub-sets. We performed the auto MRE based bootstrapping test on our set of 100 ML trees and found that convergence was reached after 50 trees.
- iii. Finally, we assessed the variance of transition rate estimates as the tree set size increases from 1 tree to 100 trees. As more trees are added, we expect the variance of estimated parameters to initially increase. However, when a sufficient number of trees have been sampled, the estimated variance should converge to the true variance, as shown in the plot below (Figure 1; corresponding to the new Supplementary Figure 22).

Figure 1. (Supplementary Figure 22 in manuscript). The variance of estimated parameters as an increasing number of trees are sampled. Trees were added in randomized order, in sets of four. Only six parameters are shown here for clarity, but the rest also show similar trends.

Though I'm sure the Authors have done a good job in documenting diversity at the habitats they sampled for long read sequencing, there may still be groups undersampled in their phylogeny. In the discussion, it would be interesting to mention if undersampling exists in the dataset and how, if at all this might have influenced rate estimations and ancestral state reconstructions.

–Response: The reviewer raises a good point. Known eukaryotic groups *not* covered by the long-read sequences are listed in Supplementary Table 2; most of the missing groups represent large, multicellular organisms, or protists found in anoxic, or other extreme environments not sampled in this study. The reviewer brings up two aspects here that can be affected by missing these and perhaps other groups, and we discuss them below.

Ancestral state reconstructions. These analyses are generally most sensitive to the deepest branching lineages in the group of interest. Therefore, if new deep-branching lineages are discovered by sampling other habitats, sequencing deeper and/or use of different primers, and added to the phylogeny, different

ancestral habitats might be inferred. Accordingly, as already shown in the manuscript, a different ancestral habitat becomes more likely for both perkinsids and choanoflagellates when we add short-read environmental data (Figure 4b; lines 304-308). We now discuss the implications of undersampling in the manuscript, particularly for the inference of the ancestral habitat of LECA, as ancestral state reconstruction becomes more challenging and has greater uncertainties for the deepest nodes (lines 348-354).

*“At its deepest phylogenetic level, our analyses suggest that the earliest eukaryotes inhabited terrestrial habitats (Figure 4) and not marine habitats as often assumed (e.g.⁵⁶⁻⁵⁸). **However, one source of uncertainties in our ancestral state reconstructions is the lack of samples from other major ecosystems such as marine sediments, anoxic habitats, and other extreme habitats like hydrothermal vents and hypersaline lakes. These habitats may contain deep-branching lineages not represented in our phylogenies. Ancestral state reconstruction analyses are most sensitive to deep-branching lineages, therefore the inclusion of environmental data from more habitat types (and the discovery of more kingdom-level lineages) is crucial to confirm the inferences presented here.**”*

Rate estimations. We have now carried out sensitivity analyses in response to another reviewer (Reviewer 2), who pointed out that the short-read data included in our manuscript is highly skewed towards soil and marine euphotic environments (with almost ten times as many reads than freshwater and marine aphotic habitats). We subsampled decreasing proportions of soil and marine euphotic OTUs and re-estimated transition rates in BayesTraits (details in Methods; lines 691-702). Our analyses suggest that transition rate estimates are largely robust to missing/incomplete phylogenies, with only two clades showing shifts in transition patterns after subsampling marine taxa (lines 217-221; Supplementary Figure 16).

I'm a bit skeptical with the high rate of transitions inferred for fungi. As the Authors also discuss briefly (l219), fungal spores can reside in environments in which the fungus is not able to live, including marine habitats. The Authors' analyses of the number of transition events correctly take this into account, but those of transition rates do not. I still think there might be an overestimation in fungi, given the ubiquitous occurrence of their spores. I don't know if a sensitivity analysis or merely interpreting these results with a grain of salt would solve this issue, but I think there is room for reconsideration here.

–Response: The reviewer raises a very good point for discussion by suggesting that the high transition rates of Fungi may be due to the ubiquitous occurrence of their spores. We assessed this possibility by examining how marine OTUs were distributed in our fungal phylogenies, and by discussing the transition rate heterogeneity within fungi. Altogether, we think that our results (presented in Supplementary Note 2; and shown below) indicate that indeed Fungi have the highest habitat transition rates among eukaryotes, which represents an interesting hypothesis to be further tested. We refer to the new Supplementary Note 2 on line 189, and hope that it satisfies the reviewer’s request: “*Fungi were found to have by far the highest number of transitions per unit of evolutionary change; we estimated around 90 expected transition events along a branch length of one substitution/site (but see Supplementary Note 2).*” The main arguments discussed in Supplementary Note 2 are presented below:

In Supplementary Note 2 we note that the high transition rate for fungi masks variation in transition of fungal groups. For instance, we found that, consistent with previous literature^{2,3}, only selected groups had marine representatives, such as Pezizomycotina (and in particular Dothideomycetes, Sordariomycetes), Saccharomycotina, Ustilaginomycotina, and Chytridiomycota, while groups like Glomeromycota did not have any marine representatives. When visually inspecting the phylogenies of groups like Pezizomycotina, we observed that marine and non-marine sequences are extremely closely related (with little to no genetic distance between them; Figure 2). This observation is consistent with the high transition rates estimated by our analyses. These results are also consistent with previous literature on the subject^{2,4}. For example, the genus *Malassezia* which is well-known to be linked to skin conditions such as dandruff, is also found in a range of marine habitats, from coral reefs to deep sea vents⁵. Furthermore, it is likely that the number of transition events in Fungi (as shown in Figure 2c-d in the manuscript) are underestimated, since we clustered OTUs at 97% similarity, and are thus unable to detect very recent transition events.

Figure 2. Subsection of phylogeny of Saccharomycotina with marine sequences in blue and non-marine sequences in orange.

We also assessed whether high transition rates were estimated simply because fungal spores are highly resistant. If fungal spores were indeed ubiquitous, we would see marine OTUs distributed indiscriminately across the fungal phylogeny. However, we observed that 359/414 (i.e. 86%) of marine OTUs belonged to clades previously known to be living in marine water column and sediment samples (Table 1). This result indicates that fungal spores may not occur as ubiquitously as presumed (and/or our stringent filtering of sequences removes most “contaminants”), and therefore should have little impact on inferred transition rates.

Table 1. Distribution of short-read OTUs among fungal clades based on a randomly selected phylogeny of long- and short-read sequences. All clades listed, except the last row, are known to have marine representatives.

	Short-read OTUs	Long-read OTUs	All OTUs
Agaricomycotina	15	17	32
Ustilaginomycotina	14	21	35
Pezizomycotina	32	62	94
Saccharomycotina	32	20	52
Puccinomycotina	23	21	44
Chytridiomycota	78	2	80
Cryptomycota	18	4	22
Sequences interspersed among other clades	46	9	55

Smaller comments:

l51 - strange phrase, maybe 'increasing <number of?> inferences?'

–Response: Thank you, fixed!

l159 - at this point it would be interesting to report rate parameters from a model in which qMT and qTM can take independent values (but uniform across the tree). Is there a rate asymmetry?

–Response: Please see our response to the first comment.

l161-162 - please report rate values from these analyses. Or, if the global tree proved for some reason unsuitable for analyzing rates, please detail why. Is there too much masking of rate values due to opposite patterns between clades? Is there an overall pattern emerging?

16–Response: We are not sure if we understand the reviewer’s request here, but we have now reported global transition rates obtained under the homogenous transition model ($q_{M-NM} = q_{NM-M} = 0.19$ transitions per substitution per site; line 162). Together with the clarification provided in response to the first comment, we hope that we have satisfied the reviewer’s comment.

l167 - how robust were those trees?

–Response: We have now described average SH-like support values for the clade-specific phylogenies on line 171-172 and a corresponding description in the Methods section on line 640-642.

l170-176 - a parsimony based introduction of the frequency of transitions would be much easier to interpret. Because bayestraits rates do not have clear parsimony interpretation, I think reporting the number of transitions inferred by parsimony, in each clade, would make interpretation easier.

–Response: We agree with the reviewer that parsimony is conceptually much simpler to understand. However, here we were particularly interested in the *overall speed* at which clades can transition between habitats (Fig 2a) - in other words how many transition events occur per unit of branch length. This cannot be achieved by reporting the number of transition events with parsimony, nor are such numbers easily comparable between groups (as they do not take the amount of genetic diversity into account). Furthermore, we do not think that parsimony is well suited for ancestral state reconstruction events as it does not take branch length into account, nor is there a statistical model underlying parsimony. Finally, reporting results from both Bayesian and parsimony analyses would make the text more confusing to read.

l201 - please indicate how significance was defined.

–Response: We have now changed the sentence on lines 207-208, which better reflects the result for centrohelids.

17l273 - 'rhizarians'

–Response: Thank you, fixed!

fig4b - a marking is missing for the polytomic mrca of ichtyosporeans and metazoans - on purpose?

–Response: This was a mistake and we thank the reviewer for pointing it out. We have now updated the figure.

l308 - well, I'm not sure this ms showed that the salt barrier per se shaped euk diversity. It showed that euk diversification involved several transitions across this barrier.

–Response: The reviewer is correct. We have now changed the sentence to “*We confirm that the salt barrier has been a major factor in shaping eukaryotic evolution...*”

l336 - I'm a bit skeptical with the fungal results, see above.

–Response: Please see our response to the corresponding comment.

Reviewer 2 (Remarks to the Author):

Jamy and co-authors present here a global study that assess the transition (marine and terrestrial) pattern and rate across the eukaryotic tree of life. To cover the general diversity of marine (euphotic

18and aphotic) and terrestrial (freshwater and soils) ecosystem, they have used a metabarcoding approach combining long-read (4500bp, 18S, ITS and 28S, PacBio) and short-read (450bp, V4 regions of the 18S rRNA gene, Illumina, 454, 22 published dataset). They have built a phylogenetic reconstruction tree based on the twenty-one samples that were sequenced with PacBio. Then they have included the short-reads in the phylogenetic reconstruction by using an EPA-ng placement. They have used multiple statistical approaches (Unifrac, Model test, ancestral state reconstruction) to investigate the pattern and the variation of transition rates among the ecosystem between major eukaryotic clades. The key findings of the study are 1) transition occurred in both directions in almost all major eukaryotic lineages 2) ancestral habitat reconstruction analyses suggest that the eucaryotes have first evolve in terrestrial environment. Overall, the study provides the first global evaluation of the role of the salt barrier in the evolution of the eucaryotes.

The manuscript is well written, The authors used the up to date tools of the field of research (Jamy et al., 2020) for the 18+3 samples (long-reads analyses) but also include 22 published dataset of short reads to complete the evaluation of the diversity. The description of the statistics, the rationale as well as the development of the manuscript is clear. The authors provide all the information (Raw Data, scripts, supplementary material, and references) to correctly redo the experiment and statistic. Nevertheless, I have highlighted some issues/limitations of the study that the authors must investigate and provide the necessary justifications before a potential acceptance in Nature Ecology and Evolution.

–Response: We thank the reviewer for their encouraging words. We have addressed their comments which we believe have improved the manuscript.

L371-408: Environmental samples for long-read metabarcoding and total DNA extraction

The authors used different approaches and methods to collect and analyze their samples. I understand the purpose of the study to remain at a global level, nevertheless each modification in the methodological procedure will induce potential biases in the interpretation of the results.

Different DNA extraction kit were used according to the sampling location. For example, for soils samples in rainforest soils (L383) the DNA were extracted using the Xpedition™ Soil/Fecal DNA miniprep and they have used the DNeasy PowerSoil Kit (Qiagen) for the samples from the Lake Erken. The choice of the DNA extraction kit can have an impact on the community profile that you will obtain (Santos et al., 2017).

Line 387-395: The authors combine plankton and sediment samples under the term “freshwater”. Nevertheless, the community composition between this two environments can be really different (Forster et al., 2016). I suppose that this choice was made to cover more diversity of freshwater ecosystem? Moreover, the authors choose to combine freshwater and soils ecosystems and coin it under the term “terrestrial.” Many authors (Singer et al., 2021; Bižić M. et al.) keep this two ecosystems separated. The aim of the article is to study the effect of the saline barrier, so in stead of “terrestrial” I would prefer to use the term “no marine.” This is way you can combine freshwater and soils.

Line 395: The quantity of water filtered, and the size of the mesh used to obtain the raw material are different between the different studies/location. This will also have an impact on the community profile obtain in the end.

All these biases will potentially impact the vision of the diversity. I suggest including a short paragraph to highlight, discuss and justify briefly all these elements inherent to every meta-analyses.

–Response: We thank the reviewer for raising these issues and for pointing us to several relevant references. We have now included a new sub-section in the methods section (“Dataset consistency”) where we discuss and justify the potential biases (lines 440-462). We address (1) different DNA extraction kits used, and (2) different mesh sizes used for processing freshwater and marine samples:

“Dataset consistency

The DNA from several samples (primarily non-Swedish samples) were obtained from other studies. The samples were therefore processed and DNA extracted using distinct methods, which can introduce various biases in the overall community obtained. We assessed how robust the community profiles obtained were to (1) different extraction methods, and (2) different filtration protocols.

- (1) Different commercial kits were used for extracting DNA from the environmental samples. Additionally, several extraction protocols (of two soil and seven marine samples) included extra cell-beating or cryogenic crushing steps^{79,80,82}. While it has been previously shown that DNA extraction protocols significantly affect the protist communities retrieved, most of these differences are restricted to several groups and do not overwhelm the real, biological variations between samples⁸⁴. Given the limited number of samples in this study, it is difficult to assess how much of the variation between samples can be attributed to the extraction protocol, but differences are likely to be minor.*
- (2) Freshwater and marine samples were filtered through meshes with different pore sizes. The smallest pore size did not differ in both cases (0.2-0.25 μm which enables capturing even the smallest eukaryotes such as *Ostreococcus tauri* with a diameter of 0.8 μm ⁸⁵). On the other hand, the largest pore size did differ between freshwater and marine samples: a “micro” size fraction*

(20-200 μm) was obtained for most freshwater samples, but not for most marine samples. However, we do not expect this difference to influence the communities obtained as the vast majority of marine protist diversity is captured in the smaller size fractions (with the exception of the exclusively marine Collodaria and Phaeodarea which are mostly found in the micro size fraction)⁶⁶. Overall, we are confident that the aquatic communities obtained are comparable to the soil communities in this study.”

Freshwater plankton and sediment samples were indeed combined to cover more freshwater diversity. However, we do not think it is necessary to justify this choice in the manuscript as the focus is on the salt barrier.

Finally, thanks to the reviewer, we see that the term “terrestrial” can be confusing when used to refer to both freshwater and soil habitats. We have now changed the term “terrestrial” to “non-marine” throughout the manuscript.

L500-555 Short read datasets

I noticed two main elements that will potentially influence the results and the conclusion of the study in the short read dataset processing. First the dataset of the short reads is completely unbalanced among the four studied ecosystem. In Sup Table 4, we can observe that there are about ten time more reads for the soil environment than for freshwater. I suppose that this fact is due to the limited numbers of study for freshwater environment. The same observation can be done for the marine euphotic and marine aphotic dataset. I recommend presenting basic analyses to provide the evidence that the numbers of dataset/sites/samples per ecosystems used in the study are enough in term of sequencing depth to cover the whole diversity of each of these environments.

–Response: The reviewer is correct to highlight that the number of reads are unbalanced between habitats with the most reads from soil and marine euphotic habitats. We would like to note that:

- 1) For our ancestral state reconstruction analyses, taxa were labelled as being either “marine” or “non-marine”, as the focus was on the salt-barrier. The total number of reads/samples/datasets between marine and non-marine habitats are rather similar (Supplementary Table 4). Both broad habitat types were 11-12 datasets, ~770 environmental samples, and ~100,000 reads.
- 2) The aim of including the short-read datasets was to incorporate more eukaryotic diversity. However, we do not assume that with these 22 studies, we cover all of the eukaryotic diversity in these four habitats. However, we do not assume that with these 22 studies, we cover all of the eukaryotic diversity in these four habitats. Indeed it is likely that the number—not the rate—

of transitions inferred across the phylogeny will increase as we incorporate sequences from more geographic locations, so in that respect our results are rather a baseline.

However, it is true that the skew in the sequencing efforts towards marine euphotic and soil habitats may impact the transition rates inferred. To assess the impact of this skew, we performed sensitivity analyses wherein we gradually removed an increasing number of soil and/or marine euphotic taxa from our phylogenies, and re-calculated transition rates. These results are now presented in a new Supplementary Figure (Supplementary Figure 16). We found that, with the exception of ciliates and gyristans, transition rates were largely consistent between subsets of data. In a few cases, transition rates varied slightly between data subsets, but importantly, the transition patterns (i.e. $q_{M-NM} = q_{NM-M}$, $q_{M-NM} > q_{NM-M}$ or $q_{M-NM} < q_{NM-M}$) did not change. These results indicate that transition rate estimates are robust to skews in sampling efforts. Interestingly, removing 30%-50% of marine euphotic OTUs shifted the transition pattern of ciliates from asymmetrical to symmetrical ($q_{M-NM} = q_{NM-M}$), and vice-versa for gyristans. These results are now presented in the manuscript (line 217-22). We emphasise that these results do not negate the overall results of the study.

The diversity obtain in the twenty-two datasets were obtain by using different couple of primers. This difference will preferentially amplify certain taxon as we can see in sup. Fig 7. As Dr. Daniel Vaultot (co-author of the article) have said in his last paper published in Molecular Ecology Resources: To cover a relevant and representative fraction of the protist community in a given study system, an informed primer choice is necessary, as no primer pair can target all protists equally well (Vaultot et al., 2021). The used of primers pair in the study are also unbalanced (Sup Table 3). With primers that were only present in a certain kind of ecosystems. This will lead to potentially the amplification of specific clades that remain undetected in the other environment. I would like to know if some transitions found in the study are not “just” due to a primers pair bias. The authors must provide the evidence that their results have not been impacted by this potential bias. And potentially redo their analyses by selecting only the taxa that are amplify by all the primers pairs.

–Response: We thank the reviewer for this valid criticism. First, we would like to note that while we agree that all primers are inherently biased in some ways against some groups (different groups for different primers), we respectfully disagree with the observation that what we see in Supp. Fig 7 (now Supp. Fig 8) is due to such bias. Instead, we argue that the differences between the phylogenetic placements in Supp. Fig 8 reflect natural variation between the habitats. For instance, no OTUs from soil and freshwater are phylogenetically placed in the exclusively marine group, Radiolaria. This result likely does not stem from primer bias, rather it is consistent with our knowledge of radiolarian habitat

22preference⁶. Similarly, MALVs (parasites in the dinoflagellate group) are only known in marine systems, and we observe accordingly that no OTUs from soil/freshwater are placed in that lineage (Supplementary Figure 13). Altogether, the phylogenetic placements from each habitat are in very large agreement with knowledge on eukaryotic diversity patterns⁶, and we do not observe any obvious evidence of widespread primer bias. The only primer bias we detected is that no discobans (diplonemids or euglenids) were present in any of the short-read datasets, likely because of their divergent and long SSU sequences⁷. However, none of the combinations of V4 primers amplified discoban sequences in *any* habitat, which ultimately does not impact our results.

That said, we assessed the reviewer's concern of whether the different primer sets used in each habitat lead to primer bias by doing an in-silico analysis using the PR² primers database. This analysis is now described in the methods section (line 601-619) and results presented in a new Supplementary Figure 21 (reproduced below).

“Testing for primer bias

The 22 short-read datasets selected for this study were generated using nine different primer pairs (Supplementary Table 3). As no primer pair can amplify all taxa equally well, using multiple primer sets can bias the microbial communities obtained, thereby impacting downstream analyses on habitat transitions. To assess whether the different primer sets used lead to primer bias (i.e. certain taxa being detected in one habitat, but not another), we tested each primer pair in an in-silico analysis using the PR² primers database (Supplementary Figure 21). Firstly, nearly half of the datasets (10/22) were generated using one primer pair (primer set 8; TAREuk454FWD1 and TAREukREV3), spanning all four habitats. Secondly, most of the variation in primer sets comes from freshwater studies, which have used seven different primer sets in total. When not allowing any mismatches, different primer sets displayed reduced affinities for different eukaryotic clades (for example primer set 16 matches less than 25% of rhizarian sequences in PR²). However, no habitat has been surveyed by primers all biased against the same clade. Secondly, we note that PCR amplifications do not always correspond exactly to in-silico analyses; i.e. some groups with mismatches against the primers can be amplified in the lab, and vice versa. Therefore, we also performed analyses while allowing for four mismatches (corresponding to the default error-tolerance in Cutadapt of 0.1 assuming all primers are 20 bp long). In this case, the primer pairs are largely equivalent, with the exception of excavates (which we did not analyse in this study on account of high phylogenetic uncertainty). We therefore do not expect taxa to be missed in a certain habitat due to primer bias.”

tion 4.0 International License, which permits use, sharing, give appropriate credit to the original author(s) and the e made. In the cases where the authors are anonymous, should be to 'Anonymous Referee' followed by a clear re included in the article's Creative Commons license, unless article's Creative Commons license and your intended use is o obtain permission directly from the copyright holder. To

Supplementary Figure 21 in manuscript. (On previous page). In-silico analysis of the nine primer pairs that generated the short-read data used in this study. Primer set IDs correspond to the ID numbers on the web application of PR2 primers⁹. (A) Primer set specificity when allowing no mismatches. (B) Primer set specificity when allowing four mismatches (which matches the default setting for the error-tolerance in Cutadapt, used for trimming primers, assuming primer lengths of 20 base pairs). The results show that no single habitat has been surveyed with primer pairs all biased against the same eukaryotic clade. We can therefore be reasonably confident that we do not detect (false positive) transition events due to eukaryotic lineages being detected in one environment but not in the other due to primer bias.

Minor comments:

The point “3. Global 18S-28S phylogeny” in the github depository is a confusing. I do not understand why do you have “Amorphea.list” file for example? I suggest adding more explanation as sentences like: “inspected all alignments visually by eye,” make the process difficult to duplicate.

–Response: Thank you for this comment. We have tried to clarify these points in the documentation. “Amorphea.list” has been renamed “Amorphea.root.list”, since it was used as an input file for rooting the global phylogeny (described in the README.md file). We have also clarified what we mean by sentences like “inspected alignments visually by eye”. In this particular case, visually inspecting alignments was used to choose between different alignment programmes – and none of the generated alignments were edited manually.

The computation of the percentage of identity (Fig 1, Sup Fig 3) against the reference database (PR2) is not well explained in the methods section. The long reads contain the full 18S rRNA gene, but most of the reference sequences in PR2, are truncated according to the specific primers used to target the organisms. How did the authors take this fact into account?

–Response: The reviewer is correct and we have now expanded the methods section to better explain the calculation of percentage identity against reference sequences (lines 496-502).

“We calculated sequence similarity of the OTUs against reference sequences in a custom PR2 database³⁰ (PR2-transitions³¹; see below) using two methods : (1) A global identity search was carried out using

25VSEARCH (--usearch_global and --iddef 1 ; Supplementary Figure 4). For this method, all references and OTU sequences were trimmed with the primers 3ndf and 1510R⁹⁸ using Cutadapt⁹⁹ to ensure that they spanned the same region. (2) Since not all sequences in PR² span the region between 3ndf and 1510R, or are targeted by this primer pair, we also estimated local similarity by BLASTing the 18S OTU sequences against references in the PR² database, and extracting the top hit with an alignment of at least 500 base pairs. The corresponding percentage identities are displayed in Figure 1.”

References: genus and species must be in “italic” (L.841: *Helkesimastix marina*)

–Response: Thank you for pointing this out. We have fixed this.

PR²: the “2” of PR² must be written as a superscript

–Response: Fixed! However, we did not write the 2 as superscript in “PR²_transitions” in order to match the corresponding file names of this custom database on FigShare.

Line 37: The term “Physicochemical” is related to the physical chemistry field of research. I suppose that the meaning of the sentence is more related to the physical (itemperature, light irradiation...) and the chemical (nutrient content, pH...) properties of the environment? The term “Physicochemical” is often miss used in ecological studies, I recommend separating the terms as it is related to two different concepts.

–Response: The reviewer is correct, and we have now changed the sentence.

Line 531: Specify the minimum numbers of samples where the OTU must be present.

–Response: Changed as suggested.

Fig 1: To better visualize the range of the percentage of identity, I suggest adding a line that indicate 100% of similarity.

–Response: Changed as suggested.

Fig 1 and Sup Fig 7) I recommend using the same color codes and names between the figures to allow a better comparison between the figures.

–Response: Thank you for pointing this out. We have updated the names in Supplementary Figure 7 (now Supplementary Figure 8). However, the figures already contained the same colour codes (both for habitat and for taxonomic groups), so we did not make any changes to the colours.

Sup Fig 4) “Freshwater=Yellow”

–Response: Changed as suggested. (Now Supplementary Figure 5).

Sup Fig 5) “Font size issue”

–Response: Thank you for your in-depth reading of this manuscript. We have fixed this (now Supplementary Figure 6).

Sup table 3 and 4) Add the numbers of samples/sites

–Response: Thank you for this suggestion. We have now added the number of samples.

Bižić M., Klintzsch T., Ionescu D., Hindiyeh M. Y., Günthel M., Muro-Pastor A. M., et al. Aquatic and terrestrial cyanobacteria produce methane. *Science Advances* 6: eaax5343.

Forster, D., Dunthorn, M., Mahé, F., Dolan, J.R., Audic, S., Bass, D., et al. (2016) Benthic protists: the under-charted majority. *FEMS Microbiology Ecology* 92: fiw120.

- Jamy, M., Foster, R., Barbera, P., Czech, L., Kozlov, A., Stamatakis, A., et al. (2020) Long-read metabarcoding of the eukaryotic rDNA operon to phylogenetically and taxonomically resolve environmental diversity. *Molecular Ecology Resources* 20: 429–443.
- Santos, S.S., Nunes, I., Nielsen, T.K., Jacquioid, S., Hansen, L.H., and Winding, A. (2017) Soil DNA Extraction Procedure Influences Protist 18S rRNA Gene Community Profiling Outcome. *Protist* 168: 283–293.
- Singer, D., Seppey, C.V.W., Lentendu, G., Dunthorn, M., Bass, D., Belbahri, L., et al. (2021) Protist taxonomic and functional diversity in soil, freshwater and marine ecosystems. *Environment International* 146: 106262.
- Vaulot, D., Geisen, S., Mahé, F., and Bass, D. (2021) pr2-primers: An 18S rRNA primer database for protists. *Molecular Ecology Resources* n/a:

Dr. David Singer

Reviewer #3 (Remarks to the Author):

The paper by Jamy et al. investigates habitat evolutionary transitions across the eukaryotic tree of life using a novel approach that combines long and short read metabarcoding, showing that eukaryotes might have evolved in non-saline habitats, the crossing of the salt barrier might promote diversification, and that such barrier has been crossed several hundred times. The methods overall are sound and innovative, and the conclusions seem appropriate based on the results.

I have a few comments outlined below:

The use of global for the long read data, for instance line 137, does not seem appropriate as the samples were obtained from only a handful of sites, even if they covered all major ecosystems. It is only justifiable to use the term global phylogeny when including the short read data as this really represents a global sample. Then it also becomes confusing when the authors refer to the global eukaryotic phylogeny. Do they mean the one including only long read data or the one they have added the short read data? My suggestion is to refer to it as the “long read data phylogeny” and avoid the use of global phylogeny, unless they show a large phylogeny with long + short read data, but it seems that the phylogenies including long + short read data are subtrees from the long read data phylogeny. An alternative would be to clearly specify what the term global means in global phylogeny.

28–Response: We apologise for the confusion here. The “global” phylogeny refers to the long-read phylogeny as it spans all major eukaryotic groups. The reviewer is correct that phylogenies containing long-read + short-read data are clade-specific phylogenies and thus referred to as “group-specific phylogenies” in the manuscript. We now state that the 18S-28S phylogeny in Figure 1 is referred to as the “*global long-read eukaryotic phylogeny... as it contains almost all known major eukaryotic lineages*” (line 108). We also consistently refer to the long-read and short-read trees for each eukaryotic group as “clade-specific phylogenies” in the manuscript.

Additionally, we have now added a new Supplementary Figure (Supplementary Figure 3) which shows an overview of the analyses carried out in this study and which we believe further clarifies the different phylogenies inferred.

Supplementary Figure 3. Broad overview of the analyses carried out in this study. Not all analyses are depicted (for example the comparison of long-read and short-read metabarcoding data as shown Supplementary Figures 1-2).

Where there any taxa that can be found in more than one environment? If so, I wonder how this is reflected in the analyses? When doing this kind of analyses, I often find that the character coding for some taxa might not be very straightforward, and sometimes one taxon can be coded as both states, so for example in this particular study I wonder if there were OTUs found in more than one environment and how that was dealt with.

–Response: We agree with the reviewer that character coding can be challenging. We did find several taxa in multiple environments; with many OTUs being shared between freshwater and soils, and also between marine euphotic and aphotic habitats (Supplementary Figure 6). In most of these cases, we found that the OTUs had greater “affiliation” (i.e. much greater read abundance) to one of the habitats. This result suggested that these OTUs do not represent generalists, but rather, have strong preference for one of the habitats. We furthermore reasoned that if there were highly similar sequences found in soils/freshwaters and the oceans (and this indeed was the case for several fungal OTUs), they likely represented reproductively isolated populations that had recently transitioned into a new habitat (they could also potentially be contaminants, but see lines 228-231 to see how we took this issue into account). Therefore, for ancestral state reconstruction analyses, we clustered sequences from each habitat into habitat-specific-OTUs, so that each OTU was associated only with a single habitat type. We have now clarified our reasoning in the methods section (lines 592-599) and hope that this answers the reviewer’s question.

While PATHd8 is one of the few methods that allows dating big phylogenies, it is also known that other methods outperform it (e.g., r8s and TreePL). PATHd8 has been shown to underestimate divergence times, and to not be very reliable (Smith and O’Meara, 2012. *Bioinformatics*, Tao et al. 2020, *Efficient Methods for Dating Evolutionary Divergences in The Molecular Evolutionary Clock*). I suggest to corroborate the results with TreePL (which also works for big phylogenies, see Sun et al. 2020. *Nature communications* 11, 3333) to ensure that the results, particularly the transition rates are not biased due to the estimation time method.

–Response: We thank the reviewer for this comment, and for pointing us to a relevant reference.

First, we would like to note that the transition rates (presented in Figure 2a-b) were estimated based on *undated* phylogenies (derived from the same initial global alignment). Accordingly, transition rates are described in the manuscript as transitions per substitution per site (instead of transitions per million years), and are therefore independent of the dating method. Instead, we converted phylogenies into

relative chronograms (with root age = 1) to infer whether transitions occurred early or late in the evolutionary history of each clade (Figure 3).

Second, following the reviewer's advice, we generated relative chronograms using TreePL (we opted not to use the "thorough" option to make computation times feasible), and re-estimated transition times as in Figure 3. We now describe this analysis in the Methods section (lines 721-726). Results were largely similar (Supplementary Figure 23); i.e. most transitions were detected in recent evolutionary times in most clades, with the exception of perkinsids, choanoflagellates, and centrohelids. However, that said, one difference between results was that transition events were more shifted towards recent time in the TreePL analysis. We opted to show the original Pathd8 results in the main figures for two reasons: (1) the phylogenies were dated without fossil calibrations, and (2) assuming that dinoflagellates originated 1 billion years ago⁸, the TreePL analysis suggests no transitions in dinoflagellates occurred before 100 mya, which contradicts previous studies documenting older transitions (140 mya and older)^{9,10}. But we have included the TreePL analysis as a new Supplementary Figure (Supplementary Figure 23) and hope that it satisfies the reviewer's suggestion.

Reviewer #4 (Remarks to the Author):

This manuscript presents a global phylogenetic analysis of salinity preference in eukaryotes. The authors construct a new eukaryotic phylogeny using long-read 18S-28S sequences, providing greater resolution and higher coverage than previous efforts, and then map habitats onto this tree. They find that in general, most lineages tend to be either freshwater or marine, consistent with infrequent ancient transitions, but at the same time they document many instances of more recent transitions. I find the most interesting result to be the clade-specific relative rates of transitions across different eukaryotic groups, and in particular the high rate in Fungi compared to other groups. This work will hopefully inspire future investigations of the biological traits and mechanisms underlying this variation across groups.

In general, I found this paper to be fun to read, interesting, and well written with clear figures, and in my opinion it will be of broad interest to protist biologists, evolutionary and paleobiologists, and ecologists. The new long-read dataset should be quite useful to other researchers. I think the methods are careful and generally well described, but I would like to see some mention of how sensitive the findings are to particular methodological choices. Below I offer some suggestions for clarity and for tempering some claims a bit.

31–Response: We thank the reviewer for their positive remarks and for their suggestions to improve the manuscript.

1. I am a bit skeptical of the ancestral state reconstruction going all the way back to the LECA. I would like to see these claims perhaps tempered a bit and/or a discussion of how robust these findings are to the inclusion of specific taxa, to specific topologies, and to specific model formulations.

–Response: This is a fair point. Reconstructing ancestral states can be challenging, particularly for ancient nodes. Below, we discuss point by point how ancestral state reconstructions can be impacted by various factors, and how we address these in the manuscript.

Specific model formulations. Previous studies have shown that the model of evolution used can influence the ancestral state reconstruction. One notable example from literature is Sánchez-Baracaldo et al. 2017¹¹ and the associated response articles^{12,13}, where different model formulations (“equal rates”, “symmetrical”, “unsymmetrical”) resulted in different ancestral habitats (marine or freshwater) inferred for Archaeplastida. However, ancestral state reconstruction analyses and tools have improved considerably since then. In this study, we used complex heterogeneous models (i.e. transition rates estimated separately for each major eukaryotic clade) in a Reversible Jump MCMC framework¹⁴ (please see Methods section “Analyses of habitat preference evolution”). Under this framework, the Markov chain samples the posterior distribution of different models of evolution as well as the posterior distributions of the parameters of these models. This approach is important to prevent overparameterization and because it may be impossible to test which model fits best due to the large number of possible models (for example in our case, we can have a model where qM-NM and qNM-M in Fungi are symmetrical, but are asymmetrical in Apicomplexa, and so on). Our analyses therefore integrate results over all possible model formulations, weighted by their probabilities.

Specific topologies. It is highly unlikely that the “best” ML phylogeny represents the true phylogeny of eukaryotes. We therefore performed the ancestral state reconstruction analyses on a set of 100 ML trees, which we now show sufficiently represent the plausible topological space given the dataset (see the new Supplementary Note 1; discussed above in response to R1). Furthermore, we tested the two most popular root positions of the eukaryotic phylogeny (Amorphea-sister, and Discoba-sister), which did not impact our inference of ancestral states. We therefore chose not to discuss at length the impact of topologies in the manuscript, but we would be happy to test any other root positions that the reviewer finds important for our conclusions.

Specific taxa. The ancestral habitat inferred is most sensitive to deep branching lineages. Therefore, if new marine deep-branching lineages are discovered (and such essential lineages continue to be discovered or rediscovered e.g. hemimastigotes, rhodelphids, picozoans), the ancestral habitat inferred

32might change. Additionally, as the reviewer pointed out in another comment, we have not sampled anoxic environments (such as marine sediments) which are host to several discobid lineages and other eukaryotic groups. The inclusion of these lineages can also have implications for the ancestral state reconstruction analyses. Thanks to the reviewer, we now highlight this issue of lineage discovery and knowledge of diversity in the discussion and have tempered our claims as well in the discussion (lines 348-354; new text in bold font for emphasis):

*“At its deepest phylogenetic level, our analyses suggest that the earliest eukaryotes inhabited terrestrial habitats (Figure 4) and not marine habitats as often assumed (e.g.^{56–58}). **However, we note that one unverified source of uncertainties in our ancestral state reconstruction is the lack of samples from other major ecosystems such as marine sediments, anoxic habitats, and other extreme habitats like hydrothermal vents and hypersaline lakes. These habitats may contain deep-branching lineages not represented in our phylogenies. Ancestral state reconstruction analyses are most sensitive to deep-branching lineages, therefore the inclusion of environmental data from more habitat types (and the discovery of more kingdom-level lineages) is crucial to confirm the inferences presented here.**”*

2. One might argue that the choice of environments/samples here is far from “comprehensive” (line 73) and does not capture “all major ecosystems”. I’m thinking of anoxic environments, animal-associated, deep sediments and subsurface, extremes of temp/pH/etc. How much of the known phylogenetic diversity is **not** captured by the datasets here? And does that influence the results?

–Response: The reviewer is correct that we were a bit overzealous here. We have replaced “**all** major ecosystems” with “**several** major ecosystems” on line 78. However, we still argue that our long-read dataset is phylogenetically extensive in that it spans all eukaryotic supergroups, and major divisions. As shown in Supplementary Table 2, our data contain 158 out of the 233 (~68%) eukaryotic Classes in PR², representing the largest multigene dataset to our knowledge. That said, we are aware that this long-read dataset does not represent all microbial eukaryotic diversity, which is why we incorporated short-read environmental sequences as well.

It is difficult to assess how results are influenced by the absence of existing lineages that we haven’t sampled. But we can be reasonably confident that the main results of the paper would still stand, that is: (1) eukaryotic groups vary in their tendencies to cross the salt barrier with Fungi exhibiting the highest transitions rates; (2) the direction of habitat transitions also varies between eukaryotic groups (and we now also show that transition patterns ; (3) we detect more transitions in recent time scales. The inclusion of more lineages may, however, change the ancestral habitat inferred, particularly if the missing lineages are deep branching (as discussed in the response to another comment above). This is already illustrated in the manuscript whereby the inclusion of short-read data leads to different results for choanoflagellates and

33perkinsids (Figure 4b). And as mentioned earlier, this phenomenon could also extend to the ancestral state reconstruction of LECA, for which reason we now include a discussion of the possible effects of missing lineages on the results obtained, as described above (lines 348-354).

3. Line 83-85: the PacBio data produces a similar profile for the DCM sample, but arguably not so much for the SRF and deep sample (the dot plots are log scale; the bars below don't look very similar). Perhaps discuss. Also I don't understand the bar plots in Fig S1b – would a Venn diagram showing the overlap be easier to read?

–Response: The reviewer is correct that we were a little bit too optimistic. While it is true that all the techniques generally detect the most important groups, their abundances are considerably different (except for the DCM, and mainly for MALV-I and MALV-II). However, given the fact that we are only analysing 3 samples and that the approaches are so different, we still see these results as reasonable. Having said that, we have now changed the lines 84-86 to:

“This comparison revealed that our long-range PCR assay followed by PacBio sequencing retrieved relatively similar eukaryotic community snapshots. Most groups were detected at comparable abundances, with the exception of the MALV-I group that was detected at greater abundances with the long-read approach (Supplementary Figures 1-2).”

Regarding Fig S1b, it is true that a Venn diagram would be a more direct way to show the results. However, because the shared proportion (of reads or ASVs) between PacBio and V4 amplicons is different for the 2 approaches, it is impossible to draw such a plot. Take for example the shared reads at 100% identity. While the shared reads of V4 represent 87% of the total V4 reads, in PacBio the shared fraction represents 64% of the total PacBio reads. Thus, a single area (i.e. the intersection of the Venn diagram) cannot represent these 2 numbers at the same time. However, we have made a couple of changes to hopefully make the figure clearer and satisfy the reviewer: (1) Redundant axis labels and titles have been removed; (2) We have changed the colours to emphasise the 'shared' portion.

4. I don't entirely understand the reason for assigning taxonomy to each of the 18S-23S sequences before building the new tree. Is this related to the monophyly constraints? Or is it necessary because there are few (any?) reference sequences as landmarks in the new phylogeny?

–Response: “Yes” to both questions! The global 18S-28S phylogeny (in Fig 1) does not contain any reference sequences, and we rely on the taxonomic annotation of the long-read OTUs for (i) enforcing monophyly of major eukaryotic clades, (ii) labelling clades in the phylogeny in Figure 1, and (iii) for carrying out clade-specific analyses (Figure 2).

34We apologise for the confusion. We have now added a sentence in the methods section (line 532-534): *“The taxonomic annotations of the OTUs were used downstream to label clades in the global long-read phylogeny (Figure 1) as well as to enforce monophyly of major eukaryotic lineages (see next section)”*. We have also depicted this process graphically in a new Supplementary Figure (Supplementary Figure 3), which we believe will clarify our workflow to readers.

5. By collapsing sequences into 97% OTUs, the methods are probably underestimating recent transitions. Might be worth highlighting in the discussion.

–Response: Thank you for this very good suggestion. We have now included this aspect in our discussion as follows:

“Our analyses detected at least 350 transition events (Figure 2), although this number is likely to be higher when considering: (1) our analyses likely did not detect several ancient transitions due to extinct lineages; (2) we are likely to detect more and more cases of recent transitions as we sequence samples from more diverse geographical locations (for e.g. in ²⁴); and (3) by clustering sequences into OTUs at 97% similarity, we are likely to miss transition events where the ribosomal sequences are less diverged (e.g. in ^{26,53,54}).”

6. Discussion of timing of transition events (line 319+): it seems misleading to suggest that transitions are more frequent in recent time. Aren't there more opportunities as you go forward in time (given the extant taxa we have)?

–Response: This is indeed an interesting point for discussion. We have several thoughts on the matter, outlined below.

1. If we understand correctly, the reviewer suggests that as we go forward in time, the number of taxa increases, leading to more opportunities for transition events. We respectfully disagree with this notion, as we have no reason to believe that the number of eukaryotic taxa that have existed on earth at any given time is at its highest today.
2. Even if the reviewer's suggestion is correct, we do not always observe more transition events in more recent time; for example the centrohelids, perkinsids, and choanoflagellates in Figure 3 (and as noted on line 251-253). Furthermore, it is possible that certain geological events/periods may have increased chances of habitat transitions in the past, such as the increase of sea levels

35and consequent flooding of the continents in the Cretaceous^{15,16}, which potentially brought freshwater and marine bodies in closer contact.

3. The reviewer is correct in that we will detect more transition events in recent time. This is because taxa that have recently transitioned have not had enough time to go extinct (since it is unlikely that all of these transitioned lineages will persist through geological time); this is already noted in the discussion (line 337-340). We can also frame this argument from the opposite perspective; i.e. the extinction of lineages means that we are not able to detect a proportion of ancient transition events.
4. Finally, our result is interesting because it has been suggested that most transitions across the salt-barrier occurred in ancient time¹⁷, and our investigations with a much greater amount of data show that this is in fact not the case.

7. I think the Discussion could touch upon functional traits that might influence the phylogenetic patterns described here. How might parasites/symbionts vs free living taxa differ in propensity for transitions? Photosynthetic vs heterotrophic vs mixotrophic? Nutrition, motility, dispersal ability, animal hosts, etc. Are there any examples from specific lineages that might be informative here?

–Response: Thank you for this suggestion! How functional traits influence habitat transition tendencies is certainly an extremely interesting question, however it can be difficult to draw conclusions. The phylogenetic groups studied in this manuscript usually cover a large range of trophic strategies and sizes (which can often be proportional to dispersal abilities). For instance, some ciliates can feed by osmotrophy, some are predatory, some parasitize animals, and some harbour photosynthetic algae. It can therefore be difficult to make sweeping statements based on functional traits.

Nevertheless, to try to satisfy the reviewer's request, we have modified the last paragraph of the discussion as follows (newly added sentences emboldened for emphasis):

*“The differences in habitat transition rates across eukaryotes are likely shaped by a host of complex factors. **Groups with lower dispersal ability, such as the large radiolarians and testate amoebae, will have a lower number of potential colonisers, decreasing transition opportunities. Trophic lifestyles may also play a role, with parasites and symbionts such as apicomplexans and perkinsids potentially expected to mirror the habitat transition histories of their respective hosts.** Varying salinity tolerance levels can also prevent or promote successful colonisation events. Among algae, comparative genomics showed large differences in gene content between marine and freshwater species, notably for ion transporters and other membrane proteins that likely play important roles in osmoregulation. These*

*different gene contents may be due, at least in part, to lateral gene transfers (LGT) that could facilitate successful crossing of the salt barrier, as proposed for other environmental adaptations. **Indeed the recent discovery of frequent LGT events in two of the most frequently transitioning eukaryotic groups hints towards the potential role of LGTs in habitat transitions: LGTs acquired from bacteria have recently been shown to be important in the evolution of diatoms and other photosynthetic stramenopiles, and the acquisition of accessory genes via transposable elements seems to be particularly common in ascomycetes.** However, this hypothesis awaits testing, and it is unclear how protists in general acquire the necessary genes for osmoregulation; whether it is through LGT (as has been shown for some halophilic protists), gene family expansion, or through re-wiring of existing metabolic pathways (as shown for the SAR11 bacteria). Other ecological factors also likely play a role as a colonising organism does not only need to adapt to a different salinity, but also has to adapt to the different nutrient and ion availabilities, and avoid being out-competed or preyed on by the resident community.*

Minor comments

1. Abstract last sentence: the paper doesn't directly address this. Consider modifying this sentence.

–Response: Yes that is true. We have now changed this sentence to “Overall, our findings indicate that the salt barrier has played an important role in shaping eukaryotic evolution, and provide a valuable perspective on habitat transitions in this domain of life.”

2. “environmental phylogeny” – 1st subheading in Results and Abstract line 25 – this phrase doesn't make sense. Do you mean a phylogeny based on environmental (as opposed to cultivars or type species) sequences?

–Response: We thank the reviewer for this comment. Yes that is indeed what we meant. We have changed “environmental phylogenies” to “phylogenies” in the abstract, and have changed the 1st subheading in Results to “Long-read metabarcoding to obtain a comprehensive phylogeny of environmental diversity”.

3. Line 176 – extra “and”

–Response: Thank you, fixed!

4. Methods: please clarify the pooling of samples earlier on (c.f. line 419-420). Were pico/nano size fractions also pooled?

–Response: Yes, amplicons from different size fractions were also pooled, and we have now clarified this in the text. The sentence now reads added [bolded for emphasis here]: “*Amplicons from replicates, size fractions, and different sites from the same sampling location were pooled at this stage.*”

References:

1. Pattengale, N. D., Alipour, M., Bininda-Emonds, O. R. P., Moret, B. M. E. & Stamatakis, A. How many bootstrap replicates are necessary? *J. Comput. Biol.* **17**, 337–354 (2010).
2. Richards, T. A., Jones, M. D. M., Leonard, G. & Bass, D. Marine fungi: Their ecology and molecular diversity. *Ann. Rev. Mar. Sci.* **4**, (2012).
3. Jones, E. B. G. *et al.* An online resource for marine fungi. *Fungal Divers.* 2019 961 **96**, 347–433 (2019).
4. Gladfelter, A. S., James, T. Y. & Amend, A. S. Marine fungi. *Curr. Biol.* **29**, R191–R195 (2019).
5. Amend, A. From Dandruff to Deep-Sea Vents: Malassezia-like Fungi Are Ecologically Hyper-diverse. *PLoS Pathog.* **10**, e1004277 (2014).
6. Singer, D. *et al.* Protist taxonomic and functional diversity in soil, freshwater and marine ecosystems. *Environ. Int.* **146**, (2021).
7. Kostygov, A. Y. *et al.* Euglenozoa: taxonomy, diversity and ecology, symbioses and viruses. *Open Biol.* **11**, (2021).
8. Strassert, J. F. H., Irisarri, I., Williams, T. A. & Burki, F. A molecular timescale for eukaryote evolution with implications for the origin of red algal-derived plastids. *Nat. Commun.* **12**, 1–13 (2021).
9. Žerdoner Čalasan, A., Kretschmann, J. & Gottschling, M. They are young, and they are many:

38- dating freshwater lineages in unicellular dinophytes. *Environ. Microbiol.* **21**, (2019).
10. Batten, D. J. Cretaceous freshwater dinoflagellates. *Cretac. Res.* **10**, 271–273 (1989).
 11. Sánchez-Baracaldo, P., Raven, J. A., Pisani, D. & Knoll, A. H. Early photosynthetic eukaryotes inhabited low-salinity habitats. *Proc. Natl. Acad. Sci.* **114**, E7737–E7745 (2017).
 12. Sánchez-Baracaldo, P. *et al.* Model choice requires biological insight when studying the ancestral habitat of photosynthetic eukaryotes. *Proceedings of the National Academy of Sciences of the United States of America* vol. 114 E10608–E106099 (2017).
 13. Nakov, T., Boyko, J. D., Alverson, A. J. & Beaulieu, J. M. Models with unequal transition rates favor marine origins of Cyanobacteria and photosynthetic eukaryotes. *Proceedings of the National Academy of Sciences of the United States of America* vol. 114 E10606–E10607 (2017).
 14. Green, P. J. Reversible jump Markov chain monte carlo computation and Bayesian model determination. *Biometrika* **82**, (1995).
 15. Sims, P. A., Mann, D. G. & Medlin, L. K. Evolution of the diatoms: insights from fossil, biological and molecular data. <https://doi.org/10.2216/05-22.1> **45**, 361–402 (2019).
 16. Tennant, J. P., Mannion, P. D., Upchurch, P., Sutton, M. D. & Price, G. D. Biotic and environmental dynamics through the Late Jurassic–Early Cretaceous transition: evidence for protracted faunal and ecological turnover. *Biol. Rev.* **92**, 776–814 (2017).
 17. Logares, R. *et al.* Infrequent marine–freshwater transitions in the microbial world. *Trends Microbiol.* **17**, 414–422 (2009).

Decision Letter, first revision:

6th May 2022

Dear Dr Burki,

Your manuscript entitled "Global patterns and rates of habitat transitions across the eukaryotic tree of

39life" has now been seen again by our 4 reviewers, whose comments are attached. While three of the reviewers are now satisfied with the manuscript, Reviewer 1 has some remaining concerns that will need to be addressed before we can offer publication in Nature Ecology & Evolution.

We therefore invite you to revise your manuscript again taking into account all reviewer comments. Please highlight all changes in the manuscript text file.

- * Include a "Response to reviewers" document detailing, point-by-point, how you addressed each reviewer comment. If no action was taken to address a point, you must provide a compelling argument. This response will be sent back to the reviewers along with the revised manuscript.
- * If you have not done so already please begin to revise your manuscript so that it conforms to our Article format instructions at <http://www.nature.com/natecolevol/info/final-submission>. Refer also to any guidelines provided in this letter.
- * Include a revised version of any required reporting checklist. It will be available to referees (and, potentially, statisticians) to aid in their evaluation if the manuscript goes back for peer review. A revised checklist is essential for re-review of the paper.

[REDACTED]

Nature Ecology & Evolution is committed to improving transparency in authorship. As part of our efforts in this direction, we are now requesting that all authors identified as 'corresponding author' on published papers create and link their Open Researcher and Contributor Identifier (ORCID) with their account on the Manuscript Tracking System (MTS), prior to acceptance. ORCID helps the scientific community achieve unambiguous attribution of all scholarly contributions. You can create and link your ORCID from the home page of the MTS by clicking on 'Modify my Springer Nature account'. For more information please visit www.springernature.com/orcid.

40Please do not hesitate to contact me if you have any questions or would like to discuss these revisions further.

[REDACTED]

Reviewers' comments:

Reviewer #1 (Remarks to the Author):

Jamy et al have improved their manuscript significantly, addressing most of the points I raised in my previous review. There are two remaining issues which I think deserve some attention, the first is the presentation of model tests and specifically how rates were configured in the analysis. The second is the interpretation of fungal results. I remain skeptical as to whether the high transition rates between marine and non-marine habitats inferred by the Authors reflect real lifestyle transitions or are simply the result of metabolically inactive spores residing in a marine environment and being picked up by metagenomic approaches.

Phylogenetic model tests

I think there is a lack of clarity here. Initially I thought the Authors' 'homogeneous model' is a model in which both rates are equal. Supplementary Figure 10 suggests this is the case indeed (both rate values are ~ 0.19). This is usually referred to as an equal rates model and is a counterpart of the 'all rates different' ARD model, in which forward and reverse transition rates can take different values. However, in the methods the model is circumscribed as a different model (qm-nm and qnm-m remain constant throughout the global eukaryotic tree). I'm not sure anymore what model was actually being used. I interpret the text as an equal rates model with a single regime across the tree being compared to an equal rates model with multiple rate regimes across the tree. In any case, my initial request referred to the 'ARD' model in which forward and reverse rates can take different values. Such models usually provide better fit to the data which can be tested formally (e.g. likelihood ratio tests); should be checked on this dataset too. This is not the central analysis in the paper, but should be clarified before publication.

Fungi

I don't fully agree with the Authors' argumentation on marine fungi. This is a highly debated field in mycology and I will respect the Authors' opinion in any case; but I think the topic deserves a little more discussion and more cautious interpretation in the text. Therefore I'm offering some alternative interpretations of the Authors' data here, which they might consider during their revision.

- The Authors mention that marine and non-marine sequences in fungi are extremely closely related, often identical. To me, this argues for contamination from spores of ubiquitous filamentous fungi,

41rather than distinct marine and non-marine taxa. If there were marine taxa, then genetic distances would be larger.

- Glomeromycota might not be frequently encountered in marine environments simply because they have big spores (~0.1mm) which do not disperse as far as microscopic spores of filamentous fungi or yeast cells. All other clades seem to be represented roughly in proportion to their diversities (based on the table in Suppl. Note 2).

- *Malassezia*, which the Authors bring up as an example, might, in fact, be a prime example of human contamination: cruising ships dump wastewater to the ocean, which contains dandruff and fungal particles from humans.

Minor comments:

Suppl. Fig 20 - which colour denotes which rate?

1250-251 - most transitions being closer to present could also argue for detecting metabolically inactive spores.

Reviewer #2 (Remarks to the Author):

Mahwash Jamy and coauthors present here a revised version of their manuscript; "Global patterns and rates of habitat transitions across the eukaryotic tree of life". As a reviewer of the first submission, I have carefully checked the validity and justification provided by the authors regarding each point raised by the reviewers. I am now satisfied and convinced by the modifications and the new analyses. Therefore, I accept the manuscript for publication in *Nature Ecology and Evolution*.

Minor comments:

A careful proofreading will have to be carried out to eliminate the last errors of formats and typos.

Here is what I noticed:

Line 455) Adapt the format as before by adding (2)

Line 468) Strange "gene⁸⁷" check your latex script.

Reference: Check and correct format citation i.e., *L973 Plasmodiophora brassicae* in italic

Reviewer #3 (Remarks to the Author):

The authors have adequately addressed the comments.

Reviewer #4 (Remarks to the Author):

I appreciate the authors' very thoughtful and constructive response to all reviewers. I think the revised manuscript is even more thought-provoking and fascinating to read. I congratulate the authors on a beautiful piece of work.

42*****END*****

Author Rebuttal, first revision:

Reviewer 1 (Remarks to the Author):

Jamy et al have improved their manuscript significantly, addressing most of the points I raised in my previous review. There are two remaining issues which I think deserve some attention, the first is the presentation of model tests and specifically how rates were configured in the analysis. The second is the interpretation of fungal results. I remain skeptical as to whether the high transition rates between marine and non-marine habitats inferred by the Authors reflect real lifestyle transitions or are simply the result of metabolically inactive spores residing in a marine environment and being picked up by metagenomic approaches.

–Response: We thank the reviewer for their new comments and address the remaining concerns below.

Phylogenetic model tests

I think there is a lack of clarity here. Initially I thought the Authors’ ‘homogeneous model’ is a model in which both rates are equal. Supplementary Figure 10 suggests this is the case indeed (both rate values are ~0.19). This is usually referred to as an equal rates model and is a counterpart of the ‘all rates different’ ARD model, in which forward and reverse transition rates can take different values. However, in the methods the model is circumscribed as a different model (qm-nm and qnm-m remain constant throughout the global eukaryotic tree). I’m not sure anymore what model was actually being used. I interpret the text as an equal rates model with a single regime across the tree being compared to an equal rates model with multiple rate regimes across the tree. In any case, my initial request referred to the ‘ARD’ model in which forward and reverse rates can take different values. Such models usually provide better fit to the data which can be tested

formally (e.g. likelihood ratio tests); should be checked on this dataset too. This is not the central analysis in the paper, but should be clarified before publication.

–Response: We apologise for the continued confusion. Below, we clarify the term 'homogeneous model' and explain how a reversible-jump MCMC framework tests *all* model formulations (equal rates; all rates different, ARD) simultaneously. We have made minor edits to Supplementary Figure 10 and the methods section to improve clarity for all readers.

The homogenous model indeed refers to a single rate regime across the global eukaryotic tree. However, no restrictions are placed on the values of q_{NM-M} and q_{M-NM} . Rather, we use a reversible jump MCMC framework which simultaneously tests an equal rates model ($q_{NM-M} = q_{M-NM}$) and ARD model ($q_{NM-M} \neq q_{M-NM}$), as shown in Supplementary Figure 10A (for more details, we refer the reviewer to the article by Pagel and Meade, 2006¹ and the BayesTraits3 documentation²). The Markov chain visits each model in proportion to its posterior probability (i.e. an equal rates model will be sampled more frequently if it has a higher posterior probability than an ARD model). In the case of the homogenous/single-rate regime, an equal rates model formulation indeed presents a better fit, and is therefore sampled more frequently, as shown in Supplementary Figure 10B and as described in the main text (“transitions from marine to non-marine habitats *were found to be* just as likely as the opposite direction across the tree...”; line 161-162). This is now also described in the legend for Supplementary Figure 10. Similarly, the heterogeneous model under the reversible jump MCMC framework has no restrictions placed on parameter values but in this case has multiple, clade specific rate regimes across the phylogeny.

We have now edited the methods section to use the same terminology as in the main text (changed/new text in bold for emphasis). We apologise for previous wording, which we realise could be confusing. The new section now reads:

“To investigate whether transition rates vary between major eukaryotic clades, we compared a **homogeneous** model (q_{M-NM} and q_{NM-M} remain constant throughout the global eukaryotic tree; **i.e. a single rate regime**) against a **heterogeneous** model (q_{M-NM} and q_{NM-M} estimated separately for each major eukaryotic clade; **i.e. multiple rate regimes**) on the global eukaryotic phylogeny. These models were compared using MCMC analyses in BayesTraits v3.0.2^{123,124} in a reversible-jump framework in order to avoid over-parameterization¹²⁵. Briefly, under this framework, the Markov chain samples the posterior distribution of different models of evolution as well as the posterior distributions of the parameters of these models; **i.e. no parameter restrictions were applied, and the Markov chain simultaneously tested “equal rates” and**

44“unequal rate” models, thereby integrating results over all possible model formulations weighted by their probabilities. Following the analysis in ¹²⁶, we used 50 stones and a chain length of 5,000 to obtain marginal likelihood for each model using stepping stone method¹²⁷, and a Log Bayes Factor (2 * difference of log marginal likelihoods) of 10 or more was used to favour the **heterogeneous** model over the **homogeneous** model.”

Fungi

I don't fully agree with the Authors' argumentation on marine fungi. This is a highly debated field in mycology and I will respect the Authors' opinion in any case; but I think the topic deserves a little more discussion and more cautious interpretation in the text. Therefore I'm offering some alternative interpretations of the Authors' data here, which they might consider during their revision.

–Response: We agree that a healthy scepticism is warranted here, and thank the reviewer for informing us that they will respect our interpretation. Below we respond to each point raised and outline several arguments for why we think most marine fungi detected are not just contaminants. We have also added a new section in Supplementary Note 2 (“**Evidence indicating that marine fungi are metabolically active**”) where we review evidence indicating that marine fungi are metabolically active.

- The Authors mention that marine and non-marine sequences in fungi are extremely closely related, often identical. To me, this argues for contamination from spores of ubiquitous filamentous fungi, rather than distinct marine and non-marine taxa. If there were marine taxa, then genetic distances would be larger.

–Response: While it is true that the marine and non-marine fungal sequences are very closely related, they are often not identical even based on 18S-28S sequences (i.e. more conserved than the ITS; Supplementary Note 2, Fig 1). It is likely that including the ITS region would reveal further genetic differences between marine and non-marine fungal taxa.

Moreover, we do not necessarily agree that closely related sequences equate contamination (for example, multiple cases of different morpho-species in marine and freshwater habitats with identical or near-identical ribosomal DNA sequences have been documented^{3,4}). It is also possible that several fungal lineages may be generalists (as noted in the main text on line 362), able to survive, grow and reproduce in vastly different niches, which would then result in minimal genetic distances between marine and non-marine sequences. For instance, many aquatic hyphomycetes thought exclusively to be saprotrophs in freshwater streams, have also been reported to be endobionts of land plants⁵⁻⁷; the fungus *Aspergillus flavus* is an opportunistic pathogen of land animals and plants, and is also commonly isolated from marine sponges and corals⁸; the genus *Cladosporium* is also found ubiquitously, from decaying vegetation, man-made environments, hypersaline habitats, among many others⁹.

- Glomeromycota might not be frequently encountered in marine environments simply because they have big spores (~0.1mm) which do not disperse as far as microscopic spores of filamentous fungi or yeast cells. All other clades seem to be represented roughly in proportion to their diversities (based on the table in Suppl. Note 2).

–Response: The reviewer is correct that dispersal ability is likely one of the many factors that determine whether or not organisms are able to successfully colonise another habitat (this is noted in the discussion; line 368-369). However, it is difficult to say which factor is the most important determinant in this case. For instance, another alternative explanation for the lack of Glomeromycota in marine habitats is that almost all taxa in this clade are obligate symbionts of plant roots. On the other hand, the marine fungi detected in this study and previous studies are mostly saprotrophic (adept at living on different surfaces), or parasitic/symbiotic (infecting diatoms, dinoflagellates, and others)¹⁰⁻¹².

Furthermore, several lines of evidence suggest that marine fungi are true marine residents, contributing to biogeochemical cycles. First, RNA sequencing of the 18S gene recovers fungi as one of the most dominant groups in marine sediment communities^{13,14}, indicating that fungi are metabolically active, and not inactive resting spores. Second, marine fungal communities have been found to be structured by habitat, temperature, salinity, and other environmental factors^{12,15,16}. For instance, saprotrophic fungi have low abundance and diversity in upper marine water column

environments (which are often nutrient poor and dominated by free floating or swimming photosynthetic and grazing protists)¹¹, but dominate sediments and marine snow particles in the bathypelagic^{14,17,18} where they presumably have increased niche availability. On the other hand, chytrids and cryptophytes infecting eukaryotic algae are more dominant in colder, pelagic habitats^{19,20}. Third, several studies have shown the role of marine fungi by measuring their extracellular enzyme activity²¹, the amount of carbon they assimilate from phytoplankton²², and assessing active metabolic pathways using metatranscriptomics^{23,24}.

- *Malassezia*, which the Authors bring up as an example, might, in fact, be a prime example of human contamination: cruising ships dump wastewater to the ocean, which contains dandruff and fungal particles from humans.

–Response: The reviewer is correct that there is a great risk of contamination (also during lab protocols) in metabarcoding studies. However, as noted above, there are multiple lines of evidence which suggest that marine *Malassezia* are actively living in the oceans. (1) *Malassezia* cDNA has been detected in marine sediments with stringent protocols ruling out the possibility of lab contamination^{13,25}; (2) Metatranscriptomics revealed *Malassezia* to be actively transcribing in coral associated communities²⁶; and (3) this fungus is highly abundant in environments far removed from humans such as deep sea vents²⁷, marine seafloor²⁸, Antarctic soils²⁹ etc.

Minor comments:

Suppl. Fig 20 - which colour denotes which rate?

–Response: Thank you for pointing this out. We have now added this information in the figure legend.

1250-251 - most transitions being closer to present could also argue for detecting metabolically inactive spores.

–Response: Please see our response to the comments above. We would like to add that in Supplementary Note 2, we acknowledge that some the marine fungal sequences could indeed be by contaminants.

Reviewer #2 (Remarks to the Author):

Mahwash Jamy and coauthors present here a revised version of their manuscript; “Global patterns and rates of habitat transitions across the eukaryotic tree of life”. As a reviewer of the first submission, I have carefully checked the validity and justification provide by the authors regarding each point raise by the reviewers. I am now satisfied and convinced by the modifications and the new analyses. Therefore. I accept the manuscript for publication in Nature Ecology and Evolution.

Minor comments:

A careful proofreading will have to be carried out to eliminate the last errors of formats and typos.

Here is what I noticed:

Line 455) Adapt the format as before by adding (2)

–Response: We are not sure what the reviewer means here. However, we have proof read the manuscript and fixed several inconsistencies.

Line 468) Strange “gene⁸⁷” check your latex script.

–Response: Thank you, fixed!

Reference: Check and correct format citation i.e., *L973 Plasmodiophora brassicae* in italic

–Response: Thank you, fixed!

Reviewer #3 (Remarks to the Author):

The authors have adequately addressed the comments.

–Response: Thank you.

Reviewer #4 (Remarks to the Author):

I appreciate the authors' very thoughtful and constructive response to all reviewers. I think the revised manuscript is even more thought-provoking and fascinating to read. I congratulate the authors on a beautiful piece of work.

–Response: Thank you very much!

Decision Letter, second revision:

26th May 2022

Dear Dr. Burki,

Thank you for submitting your revised manuscript "Global patterns and rates of habitat transitions across the eukaryotic tree of life" (NATECOLEVOL-211115093B). It has now been seen again by the original reviewers and their comments are below. The reviewers find that the paper has improved in revision, and therefore we'll be happy in principle to publish it in Nature Ecology & Evolution, pending minor revisions to satisfy the reviewers' final requests and to comply with our editorial and formatting guidelines.

49[REDACTED]

Reviewer #1 (Remarks to the Author):

The Authors have provided an elaborate answer to my questions, which I'm happy to accept.

Our ref: NATECOLEVOL-211115093B

8th June 2022

Dear Dr. Burki,

Thank you for your patience as we've prepared the guidelines for final submission of your Nature Ecology & Evolution manuscript, "Global patterns and rates of habitat transitions across the eukaryotic tree of life" (NATECOLEVOL-211115093B). Please carefully follow the step-by-step instructions provided in the attached file, and add a response in each row of the table to indicate the changes that you have made. Please also check and comment on any additional marked-up edits we have proposed within the text. Ensuring that each point is addressed will help to ensure that your revised manuscript can be swiftly handed over to our production team.

****We would like to start working on your revised paper, with all of the requested files and forms, as soon as possible (preferably within two weeks). Please get in contact with us immediately if you anticipate it taking more than two weeks to submit these revised files.****

In recognition of the time and expertise our reviewers provide to Nature Ecology & Evolution's editorial process, we would like to formally acknowledge their contribution to the external peer review of your manuscript entitled "Global patterns and rates of habitat transitions across the eukaryotic tree of life". For those reviewers who give their assent, we will be publishing their names alongside the published article.

50Nature Ecology & Evolution offers a Transparent Peer Review option for new original research manuscripts submitted after December 1st, 2019. As part of this initiative, we encourage our authors to support increased transparency into the peer review process by agreeing to have the reviewer comments, author rebuttal letters, and editorial decision letters published as a Supplementary item. When you submit your final files please clearly state in your cover letter whether or not you would like to participate in this initiative. Please note that failure to state your preference will result in delays in accepting your manuscript for publication.

Cover suggestions

As you prepare your final files we encourage you to consider whether you have any images or illustrations that may be appropriate for use on the cover of Nature Ecology & Evolution.

Nature Ecology & Evolution has now transitioned to a unified Rights Collection system which will allow our Author Services team to quickly and easily collect the rights and permissions required to publish your work. Approximately 10 days after your paper is formally accepted, you will receive an email in providing you with a link to complete the grant of rights. If your paper is eligible for Open Access, our Author Services team will also be in touch regarding any additional information that may be required to arrange payment for your article.

Please note that *Nature Ecology & Evolution* is a Transformative Journal (TJ). Authors may publish their research with us through the traditional subscription access route or make their paper immediately open access through payment of an article-processing charge (APC). Authors will not be required to make a final decision about access to their article until it has been accepted. [Find out more about Transformative Journals](https://www.springernature.com/gp/open-research/transformative-journals)

Authors may need to take specific actions to achieve [compliance](https://www.springernature.com/gp/open-research/funding/policy-compliance-faqs) with funder and institutional open access mandates. If your research is supported by a funder that requires immediate open access (e.g. according to [a](https://www.springernature.com/gp/open-research/funding/policy-compliance-faqs)

51[Plan S principles](https://www.springernature.com/gp/open-research/plan-s-compliance)) then you should select the gold OA route, and we will direct you to the compliant route where possible. For authors selecting the subscription publication route, the journal's standard licensing terms will need to be accepted, including <https://www.nature.com/nature-portfolio/editorial-policies/self-archiving-and-license-to-publish>. Those licensing terms will supersede any other terms that the author or any third party may assert apply to any version of the manuscript.

[REDACTED]

[REDACTED]

Reviewer #1:

Remarks to the Author:

The Authors have provided an elaborate answer to my questions, which I'm happy to accept.

Final Decision Letter:

23rd June 2022

Dear Dr Burki,

We are pleased to inform you that your Article entitled "Global patterns and rates of habitat transitions across the eukaryotic tree of life", has now been accepted for publication in Nature Ecology & Evolution.

Over the next few weeks, your paper will be copyedited to ensure that it conforms to Nature Ecology and Evolution style. Once your paper is typeset, you will receive an email with a link to choose the appropriate publishing options for your paper and our Author Services team will be in touch regarding any additional information that may be required

52After the grant of rights is completed, you will receive a link to your electronic proof via email with a request to make any corrections within 48 hours. If, when you receive your proof, you cannot meet this deadline, please inform us at rjsproduction@springernature.com immediately.

You will not receive your proofs until the publishing agreement has been received through our system

Due to the importance of these deadlines, we ask you please us know now whether you will be difficult to contact over the next month. If this is the case, we ask you provide us with the contact information (email, phone and fax) of someone who will be able to check the proofs on your behalf, and who will be available to address any last-minute problems . Once your paper has been scheduled for online publication, the Nature press office will be in touch to confirm the details.

Acceptance of your manuscript is conditional on all authors' agreement with our publication policies (see www.nature.com/authors/policies/index.html). In particular your manuscript must not be published elsewhere and there must be no announcement of the work to any media outlet until the publication date (the day on which it is uploaded onto our web site).

Please note that *Nature Ecology & Evolution* is a Transformative Journal (TJ). Authors may publish their research with us through the traditional subscription access route or make their paper immediately open access through payment of an article-processing charge (APC). Authors will not be required to make a final decision about access to their article until it has been accepted. [Find out more about Transformative Journals](https://www.springernature.com/gp/open-research/transformative-journals)

Authors may need to take specific actions to achieve [compliance with funder and institutional open access mandates](https://www.springernature.com/gp/open-research/funding/policy-compliance-faqs). If your research is supported by a funder that requires immediate open access (e.g. according to [Plan S principles](https://www.springernature.com/gp/open-research/plan-s-compliance)) then you should select the gold OA route, and we will direct you to the compliant route where possible. For authors selecting the subscription publication route, the journal's standard licensing terms will need to be accepted, including [those licensing terms will supersede any other terms that the author or any third party may assert apply to any version of the manuscript](https://www.nature.com/nature-portfolio/editorial-policies/self-archiving-and-license-to-publish).

An online order form for reprints of your paper is available at http://www.springernature.com/reprints

<https://www.nature.com/reprints/author-reprints.html>><https://www.nature.com/reprints/author-reprints.html>. All co-authors, authors' institutions and authors' funding agencies can order reprints using the form appropriate to their geographical region.

We welcome the submission of potential cover material (including a short caption of around 40 words) related to your manuscript; suggestions should be sent to Nature Ecology & Evolution as electronic files (the image should be 300 dpi at 210 x 297 mm in either TIFF or JPEG format). Please note that such pictures should be selected more for their aesthetic appeal than for their scientific content, and that colour images work better than black and white or grayscale images. Please do not try to design a cover with the Nature Ecology & Evolution logo etc., and please do not submit composites of images related to your work. I am sure you will understand that we cannot make any promise as to whether any of your suggestions might be selected for the cover of the journal.

You can generate the link yourself when you receive your article DOI by entering it here: <http://authors.springernature.com/share>http://authors.springernature.com/share.

[REDACTED]

P.S. Click on the following link if you would like to recommend Nature Ecology & Evolution to your librarian <http://www.nature.com/subscriptions/recommend.html#forms>

** Visit the Springer Nature Editorial and Publishing website at http://editorial-jobs.springernature.com?utm_source=ejp_NEcoE_email&utm_medium=ejp_NEcoE_email&utm_campaign=ejp_NEcoE>[www.springernature.com/editorial-and-publishing-jobs](http://editorial-jobs.springernature.com?utm_source=ejp_NEcoE_email&utm_medium=ejp_NEcoE_email&utm_campaign=ejp_NEcoE) for more information about our career opportunities. If you have any questions please click [here](mailto:editorial.publishing.jobs@springernature.com).**